



# Evaluating and Improving the Community Land Model's Sensitivity to Land Cover

Ronny Meier[1], Edouard L. Davin[1], Quentin Lejeune[1,2], Mathias Hauser[1], Yan Li[3], Brecht Martens[4], Natalie M. Schultz[5], Shannon Sterling[6], and Wim Thiery[1,7]

[1]ETH Zuerich, Department of Environmental Systems Science, Universitaetstrasse 16, 8092 Zuerich, Switzerland
[2]Now at: Climate Analytics, Ritterstrasse 3, 10969 Berlin, Germany
[3]University of Illinois at Urbana-Champaign, Department of Natural Resource and Environmental Sciences, 1102 South Goodwin Avenue, Urbana IL 61801, USA
[4]Ghent University, Department of Forest and Water Management, Coupure links 653, 9000 Ghent, Belgium
[5]Yale University, School of Forestry and Environmental Studies, 195 Prospect Street, New Haven CT 06511, USA
[6]Dalhousie University, Department fo Earth Sciences, 1459 Oxford Street, Halifax NS B3H 4R2, Canada
[7]Vrije Universiteit Brussel, Department of Hydrology and Hydraulic Engineering, Pleinlaan 2, 1050 Brussels, Belgium

*Correspondence to:* Ronny Meier (ronny.meier@env.ethz.ch)

**Abstract.** Modelling studies have shown the importance of biogeophysical effects of deforestation on local climate conditions, but have also highlighted the lack of agreement across different models. Recently, remote sensing observations have been used to assess the contrast in albedo, evapotranspiration (ET), and land surface temperature (LST) between forest and nearby open land on a global scale. These observations provide an unprecedented opportunity to evaluate the ability of land surface

models to simulate the biogeophysical effects of forests. Here, we evaluate the representation of the difference of forest minus open land (i.e., grassland and cropland) in albedo, ET, and LST in the Community Land Model version 4.5 (CLM4.5) using various remote sensing and in-situ data sources. To extract the local sensitivity to land cover we analyze plant functional type level output from global CLM4.5 simulations, using a model configuration that attributes a separate soil column to each plant functional type. Using the separated soil column configuration, CLM4.5 is able to realistically reproduce the biogeophysical

contrast between forest and open land in terms of albedo, daily mean LST, and daily maximum LST, while the effect on daily minimum LST is not well captured by the model. Furthermore, we identify that the ET contrast between forests and open land is underestimated in CLM4.5 compared to observation-based products and even reversed in sign for some regions, even when considering uncertainties in these products. We then show that these biases can be partly alleviated by modifying several model parameters, such as the root distribution, the formulation of plant water uptake, the light limitation of photosynthesis, and the

maximum rate of carboxylation. Furthermore, the ET contrast between forest and open land needs to be better constrained by observations in order to foster convergence amongst different land surface models on the biogeophysical effects of forests. Overall, this study demonstrates the potential of comparing sub-grid model output to local observations to improve current land surface models' ability to simulate land cover change effects, which is a promising approach to reduce uncertainties in future assessments of land use impacts on climate.





## 1   Introduction

While the forested area has stabilized or is even increasing over Europe and North America, deforestation is still ongoing at a fast pace in some areas of South America, Africa, and south-east Asia (Huang et al., 2009; Hansen et al., 2013; Margono et al., 2014; McGrath et al., 2015). In addition, carbon sequestration by re- or afforestation has been proposed as a strategy to

mitigate anthropogenic climate change (Brown et al., 1996; Sonntag et al., 2016), making forest loss or gain likely an essential component of future climate change. Changes in forest coverage impact climate by altering both the carbon cycle (Ciais et al., 2013) and various biogeophysical properties of the land surface such as albedo, evaporative fraction and roughness length (Bonan, 2008; Pitman et al., 2009; Davin and de Noblet-Ducoudré, 2010; Li et al., 2015). However, there exist considerable inconsistencies in the representation of biogeophysical effects amongst land surface models, thus generating a need for a thor-

ough evaluation of the representation of these effects in individual models.

Model simulations indicate that the biogeophysical effects of historical deforestation have been rather small on a global scale (Davin et al., 2007; Findell et al., 2007; Davin and de Noblet-Ducoudré, 2010; Malyshev et al., 2015). However, they have likely been significant on regional and local scales, especially over areas which experienced intense deforestation rates (De Noblet-

Ducoudré et al., 2012; Kumar et al., 2013; Malyshev et al., 2015; Lejeune et al., 2017). Similarly, present-day observational data, either based on in-situ (Juang et al., 2007; Lee et al., 2011; Zhang et al., 2014; Bright et al., 2017) or remote-sensing measurements (Li et al., 2015; Alkama and Cescatti, 2016; Li et al., 2016) show that biogeophysical effects of forests can strongly influence local climate conditions. Among the different biophysical effects, the increased surface albedo (cooling effect), the alteration of the evaporative fraction (warming or cooling effect, depending on the region and season), and the lower

surface roughness causing a reduction of the turbulent heat fluxes (warming effect) have been identified as the three main drivers of the climate impact of deforestation. However, some of these biogeophysical processes are not well represented in current land surface models. In the model intercomparison projects LUCID (Land-Use and Climate, IDentification of robust impacts) and CMIP5 (Coupled Model Intercomparison Project Phase 5), it became apparent that models disagree on several aspects of the biogeophysical impacts of historical land use and land cover change (LULCC) over the mid-latitudes of the

northern hemisphere, especially regarding the impact on evapotranspiration (ET) and temperature during the warm season (De Noblet-Ducoudré et al., 2012; Kumar et al., 2013; Lejeune et al., 2017). In addition, distinct discrepancies between present-day temperature observations and the simulated historical effects of LULCC over North America were identified (Lejeune et al., 2017). This highlights the need for systematic evaluation and improvement of the representation of biogeophysical processes in land surface models.

Observing the local climatic impact of LULCC is not straightforward. When temporally comparing observational data over an area undergoing LULCC, it is difficult to disentangle the effect of the LULCC forcing from other climatic forcings (e.g., greenhouse gas forcing). To overcome this difficulty observational studies often spatially compare nearby sites of differing land cover, assuming that they receive the same atmospheric forcing (e.g., Von Randow et al., 2004; Lee et al., 2011). Hence, the




sensitivity of land surface models to land cover can be evaluated best with observational data by spatially comparing different land cover types in models. Recently, Malyshev et al. (2015) employed a new approach to assess the local impacts of LULCC in land surface models by comparing climate variables over tiles corresponding to different plant functional types (PFTs) located within the same grid cell. Since PFT tiles within the same grid cell experience exactly the same atmospheric forcing,

the resulting sub-grid deforestation signal extracted by this method achieves good comparability to local observations which contrast neighboring forest and open land sites (Lee et al., 2011; Li et al., 2015; Alkama and Cescatti, 2016; Li et al., 2016).

Here we aim to evaluate and improve the sensitivity of the Community Land Model 4.5 (CLM4.5) to land cover, using observational data of the local contrast between forest and open land (i.e., grassland and cropland). In Section 3.1 of this study,

we systematically analyze the representation of the local difference of forest minus open land in albedo, ET and land surface temperature (LST) in CLM4.5 against the newly released observational remote-sensing-based products of Li et al. (2015). The forest signal in CLM4.5 is extracted by comparing tiles corresponding to forest and open land, similar to Malyshev et al. (2015). Given the uncertainties in observation-based ET estimates, we further extend our evaluation by including data from the Global Land Evaporation Amsterdam Model (GLEAM) version 3.1a (Miralles et al., 2011; Martens et al., 2017) and the

Global ET Assembly (GETA) 2.0 (Ambrose and Sterling, 2014), which are based on remote-sensing and in-situ observations, respectively. Finally, a sensitivity experiment is presented, in order to explore the possibilities to better represent the ET impact of forests in CLM4.5 (Section 3.2).

## 2 Methods and Data

### 2.1 Model Description and Set Up

CLM is the land surface component of the Community Earth System Model (CESM), a state-of-the-art earth system model widely applied in the climate science community (Hurrell et al., 2013). CLM represents the interaction of the terrestrial ecosystem with the atmosphere by simulating fluxes of energy, water and a number of chemical species at the interface between the land and the atmosphere. The represented biogeophysical processes include absorption and reflection of both diffuse and direct solar radiation by the vegetation and soil surface, emission and absorption of longwave radiation, latent and sensible heat fluxes

from the soil and canopy, and heat transfer into the snow and soil. Sub-grid heterogeneity is taken into account in CLM by the subdivision of each land grid cell in five land cover units (glacier, wetland, vegetated, lake and urban). The vegetated land unit is further divided into 16 tiles representing different PFTs (including bare soil).

We run the latest CLM version 4.5 at 0.5°resolution for the period 1997-2010. A five-year (1997-2001) spin-up period is

30 excluded from the analysis to minimize the impact of the model initialization. The analysis of CLM4.5 therefore covers the period of 2002 to 2010 which matches well with the observation period of 2002 to 2012 of Li et al. (2015). Assuming that the feedback of the land surface to the atmosphere is of minor importance for the sub-grid contrast between forest and open land tiles, simulations are performed in offline mode using atmospheric forcing data from the CRUNCEP v4 reanalysis product



(Vivoy, 2009; Harris et al., 2014). The land cover map and vegetation state data are prescribed based on MODIS observations (Lawrence and Chase, 2007). The land cover map from the year 2000 is kept static during the entire simulation period, since no land cover change is required to retrieve a spatial contrast between forest and open land (Fig. A1). The optional carbon and nitrogen module of CLM4.5 as well as the crop- and irrigation modules are kept inactive in our simulations.

By default, all PFTs within a grid cell in CLM4.5 share a single soil column (Oleson et al., 2013), implying that all PFTs experience the same soil temperature and soil moisture (SM). Further, the surface energy balance at PFT level is closed using the ground heat flux (GHF; i.e., it is calculated as the residual of the other energy fluxes). Hence, the soil warms in case of an energy excess at the land surface and vice versa. Warmer/cooler soil in turn will result in increased/decreased sensible and latent heat fluxes away from the ground and/or increased/decreased emitted longwave radiation, thereby counteracting the initial energy imbalance. Consequently, this model architecture eventually results in near-zero daily mean GHF, once the soil temperature has adjusted to an equilibrium state with a near-zero energy imbalance. On shared soil columns (ShSCs), however, GHFs can reach unrealistically high values for individual PFTs (Fig. A2 a and c), because a common soil temperature is artificially maintained for all PFTs, which differs from their individual equilibrium states. This assumption leads to a non-zero GHF into the soil over open land PFTs and out of the soil over forest PFTs for the majority of the locations across the globe, implying a lateral subsurface heat transport from open land towards forests (Schultz et al., 2016). To resolve this issue, Schultz et al. (2016) proposed a modification of CLM4.5 which attributes a separate soil column (SeSC) to each PFT. This modification allows the soil of individual PFTs to equilibrate to a different temperature (Fig. A3) and suppresses these unrealistically high (lateral) GHFs (Fig. A2 b and d). Here, we present results from a simulation on SeSCs, called CLM‑BASE, unless stated otherwise (Table 1). We also performed a simulation on ShSCs named CLM‑DFLT. Further, we present a sensitivity experiment, named CLM‑PLUS in Section 3.2, in which we try to alleviate some of the detected biases by modifying four aspects in the parameterization of vegetation transpiration in CLM4.5: (1) a shallower root distribution for open land PFTs, (2) a dynamic plant water uptake, in which plants only access water from the 10 % of the roots where water is most easily available, (3) decreased and increased light limitation for $C_3$ plants and $C_4$ plants, respectively, and (4) adapted values in the maximum rate of carboxylation.

## 2.2 Observational Data

A MODIS-based dataset is used to evaluate the effects of forest on local climate variables in CLM4.5 (Li et al., 2015). This data set was created by applying a window searching algorithm to remote-sensing LST, albedo, and ET products from the MODerate resolution Imaging Spectroradioameter (MODIS), in order to systematically compare these variables over forest and open land on a global scale. The data of this study, hereafter referred to as MODIS, cover the period of 2002 to 2012 and were aggregated from the initial window size of $0.45° \times 0.25°$ to $1°$ spatial resolution. Hence, the similar spatial scale of the MODIS data and the CLM4.5 simulations allows for good comparability between these two data sources.

We also use two additional observation-based datasets of ET to consider uncertainties in present-day ET estimates. A number



of different global ET products are available which, however, exhibit substantial discrepancies (Mueller et al., 2011; Wang and Dickinson, 2012; Mueller et al., 2013; Michel et al., 2016; Miralles et al., 2016). In particular, the algorithm from Mu et al. (2011) used to retrieve the MODIS ET product was found to systematically underestimate ET compared to in-situ and catchment-scale observations (Michel et al., 2016; Miralles et al., 2016). In addition, algorithms used to infer ET from remote-
sensing observations make assumptions on how LC influences ET, preventing an independent identification of the influence of LULCCs on ET. We therefore complement our evaluation of the ET impact of forest in CLM4.5 with two additional data sets: GLEAM version 3.1a and GETA 2.0.

GLEAM was introduced in 2011 (Miralles et al., 2011) and revised twice, resulting in the current version 3.1 (Martens
et al., 2017). It provides ET estimates for tall canopy, bare soil and low vegetation after Priestly and Taylor (1972). Canopy interception evaporation is calculated separately using the parameterization of Gash and Stewart (1979). GLEAM uses surface radiation, surface SM, precipitation, snow water, and vegetation optical depth observations to estimate ET globally at 0.25° resolution. To maximize spatial and temporal overlap with the MODIS observations, we choose GLEAM version 3.1 (hereafter referred to as GLEAM), which incorporates reanalysis input besides satellite observations. We compare the ET esti-
mates for tall canopy and low vegetation to model output for forests and open land, respectively. Since interception loss is only estimated for tall canopy, it was fully attributed to ET from forests.

GETA 2.0 (Ambrose and Sterling, 2014) is a suite of global-scale fields of actual ET for 16 separate land cover types (LCTs), derived from in-situ measurements between 1850 and 2010. Using a linear mixed effect model with air temperature, precipi-
tation, and incoming shortwave radiation as predictors, they obtained yearly ET estimates for each of these 16 different LCTs with a global coverage and 1° spatial resolution. We then use the same land cover map employed for the CLM4.5 simulations to weigh the different LCTs in this data set and retrieve an ET value for forest and open land (see Section 2.3 for more details). Since our CLM4.5 simulations were conducted without irrigation, we did not include the GETA 2.0 irrigation layer. We refer to this data set as GETA in this study.

## 2.3   Comparison Strategy

The forest signal in CLM4.5 is extracted by comparing the area-weighted mean of the variables of interest over all tiles corresponding to forest PFTs to the area-weighted mean over the tiles corresponding to open land (i.e., grassland and cropland) PFTs, similar to Malyshev et al. (2015). As such, it becomes possible to deduce a forest signal for every model grid cell containing any forest and any open land PFT, no matter how small the fraction of the grid cell covered by these PFTs. The different
PFT tiles within a 0.5° grid cell in our CLM4.5 simulations are subject to the exact same atmospheric forcing and are hence comparable to the almost local effect of forests retrieved at a resolution of 0.45° × 0.25° in MODIS. It needs to be noted that the MODIS observations can only be retrieved under clear-sky conditions, thereby potentially impairing the comparability to our CLM4.5 data which are not filtered for clear-sky days. Nevertheless, it was decided to include cloudy days for the analysis of the CLM4.5 simulations, to preserve the comparability to studies which do not distinguish between cloudy and clear-sky





days.

12 of the 16 PFTs of CLM4.5 are attributed to either the forest or the open land class as described in Table A1. Consistent with Li et al. (2015), open land was considered the combination of grassland and cropland. Hence, bare soil as well as shrubland are excluded from our analysis. Forest and open land ET of GETA was aggregated similarly using the same LC map as in the CLM4.5 simulations, with the LCTs of GETA attributed to the different CLM4.5 PFTs as listed in Table A2. To ensure a consistent comparison with the LST data from MODIS, we derive a radiative temperature ($T_{rad}$) from the emitted longwave radiation output ($LW_{up}$) in CLM4.5 according to Stefan-Boltzmann's law (assuming that emissivity is 1 as in Eq. 4.10 of Oleson et al., 2013):

$$T_{rad} = \sqrt[4]{\frac{LW_{up}}{\sigma}} \tag{1}$$

with $\sigma$ being the Stefan-Boltzmann constant ($5.567 \times 10^{-8}\,\mathrm{W\,m^{-2}\,K^{-4}}$). Hereafter $T_{rad}$ will be referred to as LST. For the local difference of forest minus open land in albedo, ET, daily mean LST, daily maximum LST, and daily minimum LST we will use the symbols $\Delta\alpha$(f-o), $\Delta$ET(f-o), $\Delta$LST$_{avg}$(f-o), $\Delta$LST$_{max}$(f-o), and $\Delta$LST$_{min}$(f-o), respectively.

## 3   Results and Discussion

### 3.1   Evaluation of the Local Effect of Forests in CLM4.5

#### 3.1.1   Albedo

The MODIS satellite observations and CLM - BASE agree on a generally negative $\Delta\alpha$(f-o) (Fig. 1). This difference is amplified towards the poles and in wintertime due to the snow masking effect. $\Delta\alpha$(f-o) tends to be more negative in CLM - BASE than in the satellite observations in all Köppen-Geiger climate zones (Table 2). Effectively, MODIS observations show a positive $\Delta\alpha$(f-o) for some latitude-month combinations concentrated in the tropics and sub-tropics (Fig. 1); however, these differences are mostly insignificant and are likely related to uncertainties in the MODIS observations which are more sparse over these regions due to frequent cloud coverage (Li et al., 2015).

MODIS albedo retrievals tend to underestimate albedo over grass- and cropland, especially under the presence of snow, and overestimate it over forests due to the heterogeneity of land cover within pixels (Cescatti et al., 2012; Wang et al., 2014). Therefore, it is likely that the magnitude of $\Delta\alpha$(f-o) is underestimated in MODIS. Consistently with this hypothesis, in-situ observations of paired forest and open land sites support the higher $\Delta\alpha$(f-o) found in CLM - BASE (Von Randow et al., 2004; Liu et al., 2005).





In agreement with the observations described above, earth system models concordantly simulate an albedo increase due to deforestation, which is enhanced during winter at higher latitudes (De Noblet-Ducoudré et al., 2012; Kumar et al., 2013; Lejeune et al., 2017; Levine and Boos, 2017). Nevertheless, considerable discrepancies amongst different land surface models remain regarding the order of magnitude of this increase. Hence, we encourage an extension of our analysis of CLM4.5 to

the land component of other earth system models and possibly using additional observational constraints. $\Delta\alpha$(f-o) is especially uncertain at northern high-latitudes (Loranty et al., 2014). Since this variable has the highest magnitude at the very same geographic location, a focus on these areas could be highly relevant in future studies.

### 3.1.2 Evapotranspiration

All of the considered observation-based ET products indicate that annual mean $\Delta$ET(f-o) is positive in all climate zones, de-

spite considerable variations in the magnitude of this difference (Table 2). GLEAM suggests a near zero $\Delta$ET(f-o) in the arid climate zone most likely because it uses surface SM data as an input to estimate ET. Also, GLEAM exhibits positive $\Delta$ET(f-o) over forests throughout the year in the mid-latitudes, unlike MODIS which proposes a negative $\Delta$ET(f-o) during winter. Paired-site FLUXNET studies offer and additional opportunity to compare ET over forest and over open land on a point scale. Overall, they report higher ET for tropical forests (Jipp et al., 1998; Von Randow et al., 2004; Wolf et al., 2011). In the mid- and

high-latitudes a number of FLUXNET studies observe a positive $\Delta$ET(f-o) during summer, and a near-zero negative $\Delta$ET(f-o) during winter, similar to MODIS (Fig. 2; Liu et al., 2005; Stoy et al., 2006; Juang et al., 2007; Baldocchi and Ma, 2013; Vanden Broucke et al., 2015; Chen et al., 2017). On the other hand, there are also FLUXNET observations indicating a negative $\Delta$ET(f-o) in the tropics (Van der Molen et al., 2006) and in the mid-latitudes during summer (Teuling et al., 2010). Although these contradicting results highlight some uncertainties, overall the fact that the two remote sensing-based data sets considered

in this study, GETA, which is based on in-situ ET measurements, as well as most paired-site FLUXNET studies consistently exhibit positive $\Delta$ET(f-o) in most regions, gives some confidence in these observations. Nevertheless, it needs to be noted that $\Delta$ET(f-o) in GETA looks fundamentally different when the data over irrigated crops instead of data over rainfed are considered (resulting in negative $\Delta$ET(f-o) at many locations). Therefore, distinguishing irrigated from rainfed crops in future evaluations would be essential, but remains beyond the scope of this study.

$\Delta$ET(f-o) in CLM-BASE is near zero at almost all latitudes (Fig. 2), and even negative in the equatorial and arid climate zones, unlike the global ET datasets which clearly suggest positive values for this variable (except for GLEAM in the arid region). Interestingly, deciduous trees are mostly responsible for this discrepancy at latitudes below 30° (Fig. A4). In the mid-latitudes, on the other hand, both deciduous and evergreen trees show lower ET than open land during summer and higher

ET during winter, which is inconsistent with GLEAM and, even more so, inconsistent with the seasonally-varying $\Delta$ET(f-o) in MODIS. Another noteworthy result is that the SeSC configuration (i.e., CLM-BASE) appears to impair the agreement on $\Delta$ET(f-o) between CLM4.5 and the considered observations (Fig. 3). CLM-DFLT exhibits a positive $\Delta$ET(f-o) throughout the year except for the tropical dry season which is caused by deciduous broadleaf trees exhibiting lower ET than open land (Fig. A5). There are two likely reasons for the negative bias in $\Delta$ET(f-o) introduced by SeSCs. First, the implicit lateral GH



flux from open land towards forest which occurs in CLM - DFLT (Fig. A2) provides additional energy over forests for turbulent heat fluxes. This energy source/sink for forests/open land is disabled by SeSCs. Second, the lower soil temperature of forests in CLM - BASE (Fig. A3) reduces the specific humidity gradient between the soil surface and the atmosphere and hence also the absolute soil evaporation. It needs to be noted that the weaker agreement with observational data of $\Delta$ET(f-o) in CLM - BASE

than in CLM - DFLT does not necessarily imply a worse representation of the evaporative processes in CLM - BASE, but could also originate from the fact that CLM4.5 was tuned to retrieve realistic ET values on ShSCs.

To shed light on the origin of the $\Delta$ET(f-o) bias in CLM4.5, we separately analyze the three components of ET in CLM4.5: soil evaporation (including sublimation/evaporation from the snow- and water-covered fraction of the soil), canopy interception

evaporation, and vegetation transpiration (VTR). As can be seen in Fig. 4 d, there is a distinct band around the equator where soil evaporation is lower in forests than in open land. Interestingly, both the study of Chen et al. (2017) and ours show that the lower soil evaporation signal only arises for the configuration with SeSCs (data of CLM - DFLT are not presented here). Thus, lower soil evaporation around the equator in CLM - BASE is likely related to the diminution of the soil temperature and of the available energy mentioned earlier in this section. It appears reasonable that, in comparison with open land, forests have

lower soil evaporation since the forest soil surface receives less incoming solar radiation, more of the incoming precipitation is intercepted by the canopy, and the water vapour concentrations within the canopy are higher. Yet soil evaporation and canopy interception evaporation make up a much larger proportion of the total ET in CLM4.5 (31 % and 19 %) compared to GLEAM (14 % and 10 %; Martens and Miralles (2017)). It is thus possible that the strength of this effect is too large in CLM4.5. However, most ET measurement techniques cannot distinguish among the different components of ET, making it difficult to assess

which partitioning is more realistic. Anyhow, negative $\Delta$ET(f-o) values in CLM4.5 are mainly driven by the lower VTR of forests in most regions, in particular during the wet season in the tropics and sub-tropics and during summer at higher latitudes (Fig. 4 c and f). A comparison of the absolute ET values of each PFT in CLM - BASE versus the GETA data reveals that CLM - BASE generally exhibits similar ET averages for needleleaf PFTs, lower ET averages for broadleaf deciduous trees as well as crops, and higher ET averages for non-arctic grasses and broadleaf evergreen trees (Table 3). Particularly, evergreen

and deciduous tropical broadleaf trees as well as $C_4$ grass have a bias larger than $0.2\,\mathrm{mm\,day^{-1}}$ relative to GETA. The biases of these PFTs can have a large effect on the overall $\Delta$ET(f-o) as they cover a large proportion of the land surface. Similarly, CLM - BASE considerably overestimates ET compared to in-situ measurements conducted over a pasture site in the Amazon by Von Randow et al. (2004) and underestimates ET compared to the two forest sites in Alaska reported in the study of Liu et al. (2005) (Table 4).

In summary, $\Delta$ET(f-o) in CLM4.5 exhibits considerable discrepancies to the considered global ET datasets and in-situ observations. The SeSC configuration amplifies these discrepancies, which are mainly driven by the difference in VTR of forest minus open land.



### 3.1.3   Land Surface Temperature

CLM‑BASE generally captures the sign of $\Delta LST_{avg}$(f-o) and $\Delta LST_{max}$(f-o), but shows considerable discrepancies for $\Delta LST_{min}$(f-o) in the mid-latitudes, compared to MODIS (Fig. 5). CLM‑BASE and MODIS both exhibit an overall negative $\Delta LST_{avg}$(f-o), except for winter at latitudes exceeding $30°$ (Fig. 5 a, b, and c). It appears that $\Delta LST_{avg}$(f-o) in CLM‑BASE has a positive bias in the equatorial, the arid, and the snow climate zone and a negative bias in the warm temperate zone (Table 2). The results for $\Delta LST_{max}$(f-o) appear similar to those for $\Delta LST_{avg}$(f-o) overall (Fig. 5). However, the observed magnitude of $\Delta LST_{max}$(f-o) tends to be larger (Table 2). For this variable MODIS exhibits an overall cooling effect of forests in all climate zones, including the snow climate zone where the sign of the difference changes seasonally (Fig. 5 d). $\Delta LST_{max}$(f-o) in CLM‑BASE appears qualitatively similar to the MODIS observations (Fig. 5 d, e, and f) but is biased positively in all climate zones (Table 2). In contrast, daily minimum LST shows much larger discrepancies between CLM‑BASE and MODIS (Fig. 5 g, h, and i). The MODIS data indicate that forests have a weak and mostly insignificant nighttime cooling effect in the equatorial region and a significant nighttime warming in the mid-latitudes throughout the year. On the other hand, $\Delta LST_{min}$(f-o) in CLM‑BASE is similar to $\Delta LST_{avg}$(f-o) and $\Delta LST_{max}$(f-o) (Fig. 5) i.e., forests have an overall nighttime cooling effect in all climate zones except for the neutral signal in the snow climate zone (Table 2).

While this comparison suggests considerable biases of CLM, there are three factors that can impair a full quantitative comparison of the MODIS LST difference between forest and open land with our CLM4.5 simulations. (1) As for the MODIS albedo product, the LST measurements of MODIS might often retrieve a mixed signal of various land cover types, which likely dampens $\Delta LST$(f-o). (2) MODIS LST data are retrieved under clear-sky conditions only, whereas we do not mask out cloudy days in the evaluation of the CLM4.5 simulations. Whilst albedo is likely unaffected by cloud coverage, it might well be relevant for LST, as it affects the radiation budget of the land surface. (3) The overpass times of the MODIS satellite system are at 1:30 am and 1:30 pm, hence not necessarily coinciding with the daily maximum and minimum LST in CLM4.5.

Interestingly and in contrast to LST, CLM4.5 simulates a small year-round warming effect of forests on daily maximum 2 m air temperature (T2M, Fig. A6). This contradicts a number of observational studies which show that the T2M difference of forest minus open land ($\Delta T2M$(f-o)) has the same sign, but is attenuated compared to $\Delta LST$(f-o) (Li et al., 2015; Vanden Broucke et al., 2015; Alkama and Cescatti, 2016; Li et al., 2016). The fact that we ran offline simulations might explain this behaviour, because some land-atmosphere feedbacks could be missed. However, several studies report similar discrepancies of $\Delta T2M$(f-o) in CLM with observational data for online CLM simulations (De Noblet-Ducoudré et al., 2012; Kumar et al., 2013; Vanden Broucke et al., 2015; Lejeune et al., 2017), suggesting that the behaviour of $\Delta T2M$(f-o) in our simulations may not be related to the offline mode.

Besides Li et al. (2015), a number of observational studies investigated the effect of forests on LST and/or T2M. Alkama and Cescatti (2016) report similar $\Delta LST$(f-o) as Li et al. (2015), which is not surprising, as both utilize data from the MODIS





satellite system. Further, there are a number of studies that used paired-site in-situ measurements to infer the local climate impact of forests on LST and/or T2M (Lee et al., 2011; Zhang et al., 2014; Vanden Broucke et al., 2015). CLM - BASE exhibits a weaker latitudinal dependence of $\Delta T2M_{avg}$(f-o) than the observational studies of Lee et al. (2011) and Zhang et al. (2014) (e.g., slope of $0.028\,\mathrm{K/°}$ in CLM - BASE compared to $0.070\,\mathrm{K/°}$ in Lee et al. (2011)). In contrast, the latitudinal dependence

$\Delta LST_{avg}$(f-o) in CLM - BASE ($0.10\,\mathrm{K/°}$) is stronger than the latitudinal dependence of $\Delta T2M$(f-o) in Lee et al. (2011), consistent with the observation that $\Delta LST$(f-o) is amplified compared with $\Delta T2M$(f-o) (Alkama and Cescatti, 2016). For both variables the shift from a cooling to a warming effect of forests is located further north in CLM - BASE ($43°\,N$) than the shift documented in the studies using paired-site in-situ measurements ($35°\,N$).

In the past, a number of modeling studies investigated the temperature effects of deforestation, using different methodologies and land surface models. An assessment of the two model intercomparison projects CMIP5 and LUCID revealed the inability of most land surface models to reproduce the observed effect of forests on daily maximum and minimum temperatures in North America (Lejeune et al., 2017). For CLM4.5, this issue can be resolved to a large extent for $\Delta LST_{max}$(f-o) by introducing the SeSC modification proposed by Schultz et al. (2016) (Fig. A7). On the other hand, CLM - BASE still is unable

to represent the nighttime warming effect of forests in the mid-latitudes exhibited by observational data (Lee et al., 2011; Zhang et al., 2014; Vanden Broucke et al., 2015; Li et al., 2015; Alkama and Cescatti, 2016; Li et al., 2016). Further, we highlight two modeling studies that can be directly compared to our study. Malyshev et al. (2015) applied a similar comparison strategy using coupled simulations conducted with the GFDL ESM2Mb model. Their results for the canopy temperature difference between cropland and natural vegetation look similar to ours for LST, with a typical shift from a cooling effect of forests at low latitudes

to a warming effect north of roughly $45°\,N$. Nevertheless, the region with a warming effect of forests extends more towards the equator in the study of Malyshev et al. (2015). The second study we discuss here is the one of Winckler et al. (2017), investigating the contrast between forest and grassland in the MPI-ESM model. This model exhibits a similar daily mean local LST difference of forest minus grassland as CLM - BASE. Further the study of Winckler et al. (2017) revealed that extensive changes in forest fraction can potentially have substantial non-local impacts not only on LST but also on precipitation, hence

emphasizing that contrasting forests and open land locally is not sufficient to fully capture the biogeophysical impact of large-scale de- or afforestation.

Inadequate representation or omission of several processes in CLM4.5 could be the source of the discrepancies of the $\Delta LST$(f-o) variables with MODIS we observe. The biases in both $\Delta LST_{max}$(f-o) and $\Delta LST_{min}$(f-o) could be alleviated by accounting

for vegetation heat storage, a process which is currently disregarded in CLM4.5. Observations estimate diurnal vegetation heat fluxes with an amplitude of $10–20\,\mathrm{W\,m^{-2}}$ in the mid- and high-latitudes (McCaughey and Saxton, 1988; Lindroth et al., 2010; Kilinc et al., 2012) and $20–70\,\mathrm{W\,m^{-2}}$ in the tropics (Moore and Fisch, 1986; Meesters and Vugts, 1996; Dos Santos Michiles and Gielow, 2008). Fluxes of this magnitude are sufficient to considerably decrease daily maximum LST and increase daily minimum LST in forests and hence potentially resolve the discrepancies of CLM4.5 with MODIS for $\Delta LST_{max}$(f-o) and





$\Delta LST_{min}$(f-o). The positive bias in $\Delta LST_{max}$(f-o) of CLM4.5 could also be related to the negative bias in $\Delta ET$(f-o), as increasing/decreasing ET over forest/open land would result in locally lower/higher temperatures.

## 3.2 Sensitivity Experiment to Alleviate ET Biases in CLM4.5

In the previous section, the most striking discrepancies between the effect of forests in CLM‑BASE and observation-based data were found for $\Delta ET$(f-o). The main driver responsible for these differences was identified to be VTR (Fig. 4). In addition, it became apparent that the SeSC configuration impairs the $\Delta ET$(f-o) compared to the ShSC configuration (Figs. A4 and A5), despite improving $\Delta LST_{avg}$(f-o) and $\Delta LST_{max}$(f-o) (Fig. A7). Hence, in this section we aim to improve $\Delta ET$(f-o) in CLM‑BASE compared to observation-based results by testing a new parameterization of VTR in a sensitivity experiment called CLM‑PLUS. This configuration of CLM4.5 incorporates four additional modifications besides the SeSCs (Table 1):

– Shallower root distribution for grass- and cropland PFTs. CLM4.5 accounts for SM stress on transpiration through a stress function $\beta_t$, which ranges from zero (when soil moisture limitation completely suppresses VTR) to one (corresponding to no soil moisture limitation of VTR). Forests for the most part experience higher SM stress than open land in CLM‑DFLT except in the northern high-latitude winter (Fig. A8), partly caused by the similar root distribution for all PFTs but evergreen broadleaf trees (Fig. 6). In reality, considerably higher maximum rooting depths of forest than grassland and cropland are observed (Canadell et al., 1996; Fan et al., 2017). Likewise, in-situ observations in the tropics show that grassland ET decreases during dry periods, because grasses have only limited access to water reservoirs located below a depth of $2\,\mathrm{m}$ (Von Randow et al., 2004). Hence, we aim to increase SM stress of open land PFTs and reduce their ability to extract water from the lower part of the soil, by introducing a shallower root distribution for these PFTs (Fig. 6).

– Dynamic plant water uptake. Tropical forests are often observed to exhibit increased ET during dry periods, due to increased incoming shortwave radiation (Da Rocha et al., 2004; Huete et al., 2006; Saleska et al., 2007). That is, despite the upper soil being dry, tropical trees still have sufficient access to water from deeper soil layers (Jipp et al., 1998; Von Randow et al., 2004). We aim to allow a similar behaviour in CLM4.5 by introducing a dynamic plant water uptake, where plants only extract water from the 10 % of the roots with best access to SM (example in Fig. A10).

– Light limitation reduction for all $C_3$ PFTs and enhancement for $C_4$ PFTs. In CLM‑BASE ET of boreal PFTs is underestimated compared to GETA (Table 3). Since VTR of these PFTs is only weakly affected by SM stress, light limitation for $C_3$ plants is reduced. On the other hand, $C_4$ grass shows a considerable positive bias in ET, which we try to alleviate by increasing the light limitation of this PFT.

– Modified maximum rates of carboxylation ($V_{cmax}$; Table A3). This PFT-specific parameter is suitable to tune VTR, since it is not well constrained from observations and VTR in models is highly sensitive to this parameter (Bonan et al., 2011). The new values were chosen with the aim to alleviate biases relative to GETA (Table 3) and still lie well within the range of observations collected in the TRY plant trait database (Boenisch and Kattge, 2017). Additionally, the





minimum stomatal conductance of $C_4$ plants, which is by default four times larger than that of $C_3$ plants, was reduced from $40000\,\mu\mathrm{mol}\,\mathrm{m}^{-2}\,\mathrm{s}^{-1}$ to $20000\,\mu\mathrm{mol}\,\mathrm{m}^{-2}\,\mathrm{s}^{-1}$ (see Eq. 8.1 in Oleson et al. 2013).

A technical description of these modifications as well as a discussion of the effect on ET by each individual modifications is provided in Appendix A.

$\Delta\alpha$(f-o) is only marginally affected by the modifications of CLM‑PLUS compared to CLM‑BASE (Fig. 7 a). This is expected since the modifications are targeted at VTR which is not linked directly to albedo. $\Delta$ET(f-o) in CLM‑PLUS is in better agreement with observations than that in CLM‑BASE. This is mainly due to an increase of ET over forests in tropical regions, thereby alleviating the observation-contradicting sign of the forest signal in CLM‑BASE (Fig. 3). The bias in average

ET compared to GETA is smaller in CLM‑PLUS than in CLM‑BASE for all PFTs except for boreal deciduous needleleaf trees and crops (Fig. 7 f). Some discrepancies with observation-based ET products nevertheless remain. $\Delta$ET(f-o) in CLM‑PLUS is still mostly less positive compared to remote sensing-based observations and GETA, and remains of opposite sign during the warm season in the temperate regions and in a narrow band around the Equator (Fig. 3 and Fig. 7 e). Similarly, the biases of CLM‑PLUS relative to the in-situ observations of Von Randow et al. (2004) are reduced, but are not completely

eliminated (Table 4). Hence, our results call for stronger modifications of the PFT-specific representation of ET in CLM4.5. While testing new model configurations, care should be taken that the implemented modifications do not impair other features of the model, related not only to the water but also the energy and carbon budgets. Reassuringly, we find that global ET values are only weakly affected in the sensitivity experiment, with $70223\,\mathrm{km}^3\,\mathrm{yr}^{-1}$ in CLM‑BASE compared to $69023\,\mathrm{km}^3\,\mathrm{yr}^{-1}$ in CLM‑PLUS. These values correspond to averages of $1.43\,\mathrm{mm}\,\mathrm{day}^{-1}$ and $1.41\,\mathrm{mm}\,\mathrm{day}^{-1}$, respectively, which lie within the

range of estimates from surface water budgets in the order of $1.2\,\mathrm{mm}\,\mathrm{day}^{-1}$ to $1.5\,\mathrm{mm}\,\mathrm{day}^{-1}$ (Wang and Dickinson, 2012). Nevertheless, it would be desirable in future studies to evaluate the biogeochemical effects of forests in the different model configurations investigated here alongside the biogeophysical effects of forests.

As a consequence of the improved $\Delta$ET(f-o), we find that CLM‑PLUS partly alleviates the positive bias $\Delta\mathrm{LST}_{max}$(f-o)

compared to the MODIS data, especially in the equatorial climate zone (Fig. 7 b and c). This hints that a realistic representation of $\Delta$ET(f-o) is crucial for resolving the underestimated cooling effect of forests on daily maximum LST. The fact that the bias in $\Delta\mathrm{LST}_{max}$(f-o) is not completely removed in CLM‑PLUS is another indication that $\Delta$ET(f-o) is still underestimated in the model.

We acknowledge that parameter tuning might not be sufficient to remove the ET biases completely. Most values in the parameterization of photosynthesis are shared by all $C_3$ PFTs, which hinders the tuning of the ET difference amongst different PFTs. For example, the introduction of PFT-specific values for some parameters in the calculation of the leaf stomatal resistance (see Eq. 8.1 of Oleson et al., 2013) would allow for a more effective tuning of ET at PFT level. Additionally, the modifications added in CLM‑PLUS only weakly affect ET at high-latitudes (e.g., Fig. 3). Modifying the temperature dependence of

photosynthesis, a parameter which was not taken into consideration in this study, could be beneficial in those regions.



## 4 Conclusions

In this study we evaluate the representation of the local biogeophysical effects of forests in the Community Land Model 4.5 (CLM4.5), using recently published MODIS-based observations of the albedo, evapotranspiration (ET), and land surface temperature (LST) difference between forest and nearby open land. Given the uncertainties in observation-based ET estimates, we

further extend our evaluation for this variable by including data from GLEAM v3.1a and GETA 2.0. In our model evaluation we extract a local signal of forests by analyzing PFT-level model output, allowing for good comparability with the high-resolution satellite observations. Further, we use a modified version of CLM4.5 which attributes a separated soil column to each plant functional type (PFT), resulting in a more realistic sub-grid contrast between forest and open land.

Overall, the lower albedo over forests in CLM4.5 is in line with the MODIS observations. However, the albedo contrast between forest and open land is somewhat more pronounced in the model. Ground observations support the stronger albedo contrast in CLM4.5, indicating that MODIS albedo observations should be used carefully when contrasting different land cover types, as satellite observations tend to retrieve a mixed signal of various land cover types due to their limited spatial resolution. By suppressing lateral ground heat fluxes, the soil column separation considerably improved the representation of the impact

of deforestation on daily mean and maximum LST, resulting in a good agreement with the MODIS observations. Both exhibit an overall cooling effect of forests on these variables, except for winter at latitudes exceeding 30° (Fig. 5). Nevertheless, it appeared that the LST difference of forest minus open land in CLM4.5 tends to have a positive bias compared to observational studies. Also, it emerged that caution is required when comparing 2 m air temperature in CLM4.5 to observational data. This variable is only diagnostic in CLM4.5 and might not be conform with measurements, despite realistic LST values. The night-

time warming effect of forests in the mid-latitudes which emerged in a number of observational studies, is not represented by CLM4.5. This issue has been observed in other modeling studies using CLM. The biases in the daily maximum and minimum LST signal of forests might be at least partly alleviated by accounting for heat storage in the forest biomass. We therefore encourage a modification of CLM which enables the representation of canopy heat storage.

Observation-based ET estimates generally agree on higher ET over forests than open land throughout the year at low latitudes and during summer at mid- and high latitudes. This was however not represented by the CLM4.5 configuration using separated soil columns. In fact, the soil column separation impaired the ET signal of forests in CLM4.5, despite improving the LST signal of forests considerably. Hence, a complete evaluation and verification of this modification of CLM4.5 should be undertaken before including it in future versions of CLM. The ET difference of forest minus open land is mainly driven

by vegetation transpiration. Therefore a revision of the parameterization of transpiration appears necessary to achieve better comparability with observations on the ET difference of forest minus open land. We succeeded in attenuating the biases in ET and also daily mean and maximum LST in a sensitivity experiment which incorporated modifications on four aspects of the parameterization of vegetation transpiration: The root distribution, a dynamic plant water uptake instead of the current static one, the light limitation, and the maximum rate of carboxylation.





Historically the most important LULCC process, deforestation is still ongoing mainly in South America, Africa, and south-east Asia. A realistic representation of the biogeophysical effects of Land Use and Land Cover Change (LULCC) in climate models is needed as a number of observational studies revealed that they can have a considerable impact on the local climate. An ap-
5  propriate representation of the effects of LULCC is not only a feature land surface models need to have in order to understand the past and future climate, but is also a chance to achieve a more realistic simulation of processes at the land surface. As can be seen from the analysis of ET in this study, model output at PFT level can reveal model deficiencies that otherwise would have been hidden below the veil of aggregation and can thus facilitate a better understanding of the underlying processes.





## Appendix A: Sensitivity of CLM4.5 to Individual Modifications

Here we present a more detailed description and discussion of the individual modifications described in Section 3.2. In order to isolate the effect of the individual modifications three additional sensitivity experiments are presented: CLM‑ROOT, CLM‑10PER, and CLM‑LIGHT. Table A4 shows which modifications of CLM4.5 are incorporated in the different sensitivity
experiments.

### A0.1 Sensitivity to Root Distribution

In CLM4.5 ET is strongly and positively correlated to SM at most locations, indicating that SM limitation exerts a strong control on the magnitude of ET (not shown). In CLM‑DFLT, where SM is the same for all PFTs within a grid cell, forest mostly experiences higher SM stress except for the northern high-latitude winter (Fig. A8 a). Once the SeSCs are introduced
in CLM‑BASE, the differences in the SM stress are also influenced by the differences in SM, which in term are affected by the various ET rates over forest and open land. In other terms, it is possible that forests experience less SM stress than open land but only because they evaporate less water and vice versa (Fig. A8 b). We argue that the difference in the SM stress of forest minus open land in CLM‑DFLT is more representative, because it is unaffected by the ET rates of the individual PFTs in this model configuration. Under this assumption, forests are often more SM-limited than open land in CLM4.5. In contrast,
two observational studies comparing SM profiles of forest and nearby pasture sites in the Amazon reveal that forests have a considerably higher capacity to access water from the soil below a depth of $2\,\mathrm{m}$ (Jipp et al., 1998; Von Randow et al., 2004). Further, there are a number of studies reporting increased forest ET during the dry season due to the higher amount of incoming shortwave radiation, whilst the response is the opposite over pasture (Jipp et al., 1998; Da Rocha et al., 2004; Von Randow et al., 2004; Huete et al., 2006; Saleska et al., 2007). Altogether these studies indicate that forest ET should be less SM-limited
than open land ET. It is thus possible that forests experience too high and/or open land too little SM stress in CLM4.5.

CLM4.5 accounts for SM stress on VTR through a stress function $\beta_t$, which ranges from zero (when soil moisture limitation completely suppresses VTR) to one (corresponding to no SM limitation on VTR). This function is calculated according to Eq. A1 as the sum of the root fraction in each soil layer ($r_i$) multiplied by a PFT-dependent wilting factor ($w_i$). The original
root distributions in CLM4.5 were adapted from Zeng (2001) and are rather similar for all PFTs, especially for needleleaf trees, broadleaf deciduous trees, and grassland in the lower part of the soil (Fig. 6). Therefore, there is no considerable difference in the default configuration of CLM4.5 regarding the ability to extract water from the lower part of the soil between forests and open land PFTs (except for broadleaf evergreen trees). Furthermore, all tree PFTs have a less negative soil matrix potential at which the stomata are fully closed and opened than the open land ones, i.e., tree PFTs have their permanent wilting point at a
higher SM content than open land and hence use water more conservatively. In order to increase SM limitation for open land PFTs and thus reduce their ability to extract water from the lower part of the soil, we conduct a sensitivity experiment, called CLM‑ROOT, with a much shallower root distribution for open land PFTs. The new values for the root distribution factors ($r_a$ and $r_b$) are shown in Table A3 and the resulting root distribution in Fig. 6.



$$\beta_t = \sum_i w_i r_i \tag{A1}$$

The modified root distributions strongly reduce the ET of non-arctic open land PFTs, especially ET of $C_4$ grass (Table A5).

Also, the ET of grassland at the location of the pasture site in the Amazon in the study of Von Randow et al. (2004) is considerably reduced during the dry period, even overcompensating the positive bias in CLM - BASE (Table A6). On the other hand, it does not affect ET during the wet season, when ET is not SM limited. Overall, this experiment reveals that modifying the root distribution has high potential to alleviate biases of CLM4.5 in ET, except for the arctic region where likely temperature and incoming shortwave radiation are the main factors limiting VTR.

**A0.2    Sensitivity to Dynamic Plant Water Uptake**

In the tropics forests often exhibit increased ET during dry periods, due to increased light availability (Da Rocha et al., 2004; Huete et al., 2006; Saleska et al., 2007), even though the upper soil is dry, as they still have sufficient water supply from the lower part of the soil (Jipp et al., 1998; Von Randow et al., 2004). We aim to allow a similar behaviour in CLM4.5 by introducing a dynamic plant water uptake, where plants only extract water from the 10 % of the roots with the highest wilting factor

(i.e., best access to SM) for the calculation of the $\beta_t$-factor and the extraction of soil water (example in Fig. A10). The resulting model simulation, called CLM - 10PER, was conducted by adding this modification to the configuration from the CLM - ROOT experiment.

This modification generally reduces SM stress for plants and hence increases ET for all non-arctic PFTs (Table A5). Its impact

is limited for arctic PFTs where temperature and shortwave radiation are more important limiting factors of VTR than water availability. A notable improvement can be observed for tropical deciduous broadleaf trees for which average ET is increased by $0.11\,\mathrm{mm\,day^{-1}}$, thereby alleviating the negative bias compared to GETA. Furthermore, it improves the seasonal dynamics of forest ET in the tropics. With the 10 % modification forests show increased ET during the dry period at the forest site of Da Rocha et al. (2004). This is the case as trees are now less SM-limited during the dry period than in CLM - BASE, since

they have a significant fraction of their roots in the still moist lower part of the soil, allowing them to exploit the increase in incoming shortwave radiation. On the other hand, ET at the pasture site of Von Randow et al. (2004) remains largely unaffected, as grassland has only limited access to SM from the lower part of the soil due to the shallow root distribution introduced in CLM - ROOT. It hence appears that a dynamic plant water uptake could be crucial for the representation of the seasonal dynamics of ET (and possibly photosynthetic activity in general) in the tropics.

**A0.3    Sensitivity to Light Limitation**

As arctic PFTs are only weakly affected by the previously introduced modifications of SM stress as well as the maximum rate of carboxylation described in the next section, we performed a sensitivity experiment with altered light limitation, which is



called CLM - LIGHT. Since ET values are strongly negatively biased for boreal deciduous broadleaf trees and $C_3$ arctic grass (Table A5), the light limitation of photosynthesis for $C_3$ plants was lessened by increasing the factor 0.5 in Eq. 8.7 of Oleson et al. (2013) to 0.6. Because ET of $C_4$ grass exhibits a strong positive bias, their quantum efficiency was reduced from 0.05 to 0.025 $\mathrm{mol\,CO_2\,mol^{-1}}$ photon, thereby increasing their light limitation.

Altering the light limitation of photosynthesis impacts ET in all climate zones (Table A5). Its impact is strongest in the tropics and remains small in boreal regions. Of the $C_3$ PFTs tropical evergreen broadleaf trees are impacted strongest. The implemented modification alleviates the negative ET bias for evergreen broadleaf trees during the dry season but slightly increases the positive bias during the wet season, overall still leading to a further improvement of the difference between the two seasons

(Table A6). Additionally, the increased light limitation reduces ET of $C_4$ grass during the wet season similar to the observations over the grassland site in Von Randow et al. (2004). This is likely responsible for the increased ET during the dry season as well, since the reduced SM consumption during the wet season is carried over to the following dry season, therefore reducing the SM stress.

### A0.4   Sensitivity to the Maximum Rate of Carboxylation

$V_{cmax}$ appears to be a suitable parameter to tune VTR values, since it is not well constrained from observations and VTR in models is highly sensitive to this parameter (Bonan et al., 2011). In CLM4.5 the values reported by Kattge et al. (2009) are used except for tropical evergreen broadleaf trees, for which a higher value was chosen to alleviate model biases (Bonan et al., 2012; Oleson et al., 2013). In order to test the sensitivity of the PFT-specific ET values to $V_{cmax}$, we conduct a final sensitivity experiment with new values of this parameter (Table A3) in addition to the other modifications presented beforehand, with the

aim to alleviate the biases to GETA. Additionally, the minimum stomatal conductance of $C_4$ plants, which is by default four times larger than that of $C_3$ plants, was reduced from 40000 $\mathrm{\mu\,mol\,m^{-2}\,s^{-1}}$ to 20000 $\mathrm{\mu\,mol\,m^{-2}\,s^{-1}}$ (see Eq. 8.1 in Oleson et al. 2013) in this sensitivity experiment, which we call CLM - PLUS.

As already shown by Bonan et al. (2011), photosynthetic activity of $C_3$ PFTs is strongly influenced by the choice of $V_{cmax}$,

except for the boreal ones where light or temperature are more important limiting factors of photosynthesis (Tables A5 and A6). The CLM - PLUS simulation alleviates biases in ET averaged for the individual PFTs compared to GETA, in particular by reducing ET over temperate evergreen needleleaf trees, both temperate and tropical evergreen broadleaf trees, and $C_4$ grass, as well as by increasing ET of tropical deciduous broadleaf trees. The mismatch between results of CLM4.5 and the in-situ measurements of Von Randow et al. (2004) and Da Rocha et al. (2004) in the Amazon region are reduced in this new configu-

ration during the wet season, but enhanced during the dry one. As in the CLM - LIGHT experiment the reduction of $C_4$ grass ET during the wet season at the pasture site of Von Randow et al. (2004) is partly compensated by an ET increase during the dry period. Overall, ET of $C_4$ grass compares well with the mean value of GETA. The in-situ observations of Von Randow et al. (2004) on the other hand support a stronger tuning for this particular PFT in order to further reduce ET.





*Acknowledgements.* We thank Sonia Seneviratne for her comments on the analysis and are grateful for the technical support of Urs Beyerle during the course of this project. We acknowledge funding from the Swiss Federal Office for the Environment (FOEN) and from the European Union's Horizon 2020 research and innovation programme under grant agreement No 641816 (CRESCENDO).




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





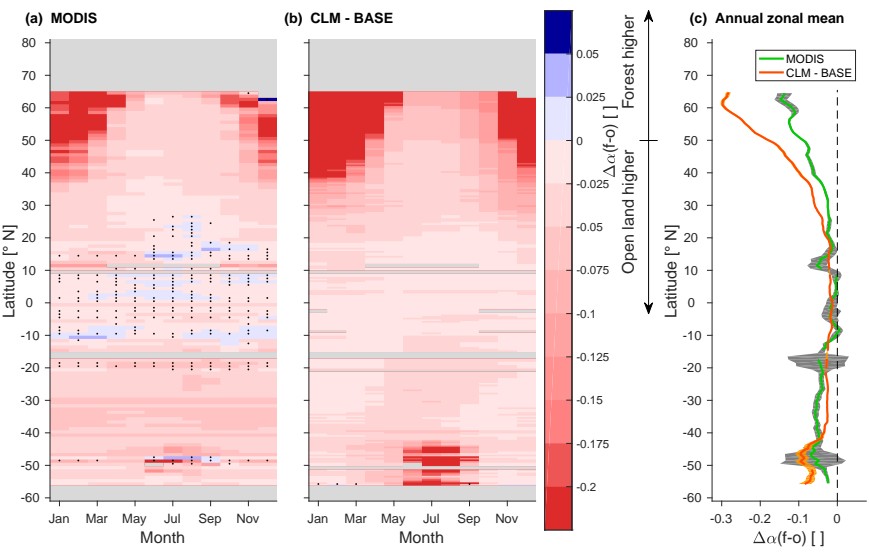

**Figure 1.** Seasonal and latitudinal variations of $\Delta\alpha$(f-o) in (a) the MODIS observations and (b) CLM - BASE. Points with a mean which is insignificantly different from zero in a two-sided t-test at 95 % confidence level are marked with a black dot. All data from the 2002-2010 analysis period corresponding to a given latitude and a given month are pooled to derive the sample set for the test. Panel (c) shows the zonal annual mean of both MODIS (in green along with its confidence interval in grey) and CLM - BASE (in red, confidence interval in orange). Note that on this subfigure results have been smoothed with a 4° latitudinally-running mean. Only grid cells containing valid data in the MODIS observations were considered for the analysis of CLM - BASE.





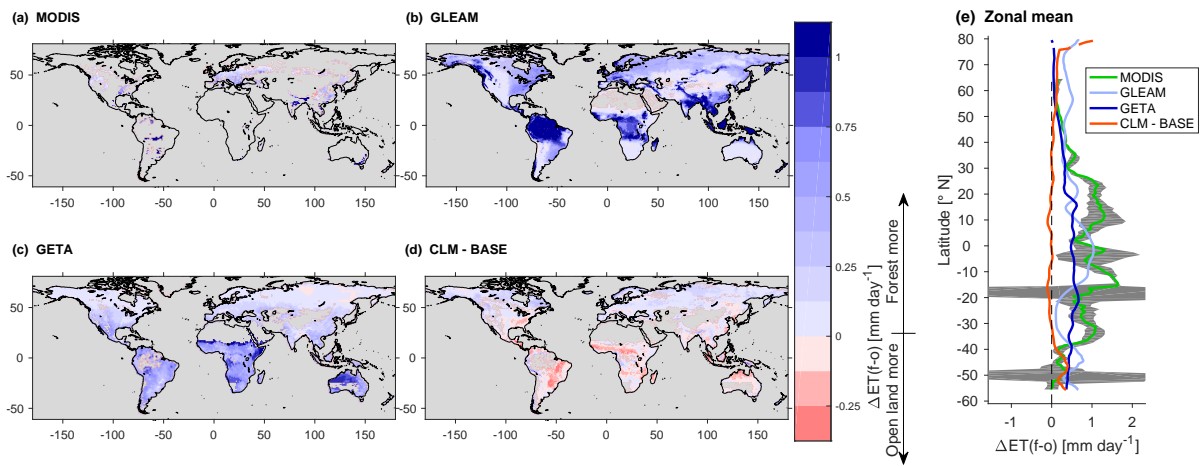

**Figure 2.** Annual mean ΔET(f-o) in (a) MODIS, (b) GLEAM, (c) GETA, and (d) CLM - BASE. Panel (e) shows the zonal mean of MODIS (in green along with its confidence interval in grey), GLEAM (blue), GETA (orange), and CLM - BASE (red). Note that on this subfigure results have been smoothed with a 4° latitdudinally-running mean.



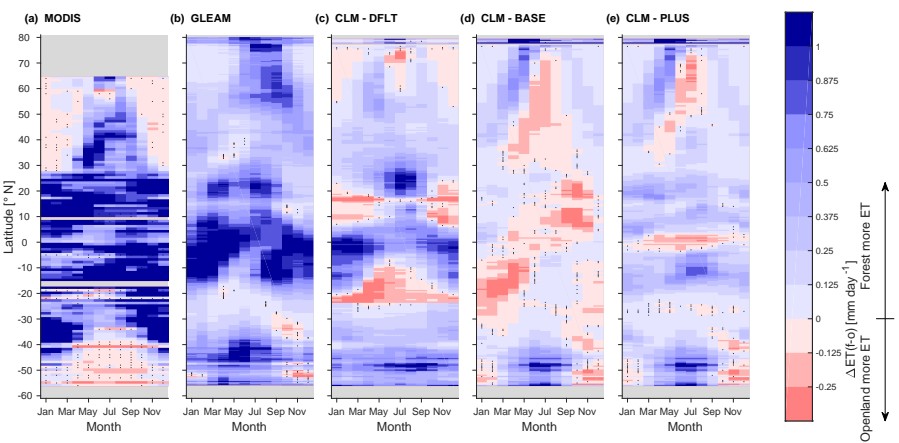

**Figure 3.** Seasonal and latitudinal variations of ΔET(f-o) in (a) the MODIS and (b) GLEAM observations, (c) CLM - DFLT, (d) CLM - BASE, and (e) CLM - PLUS.





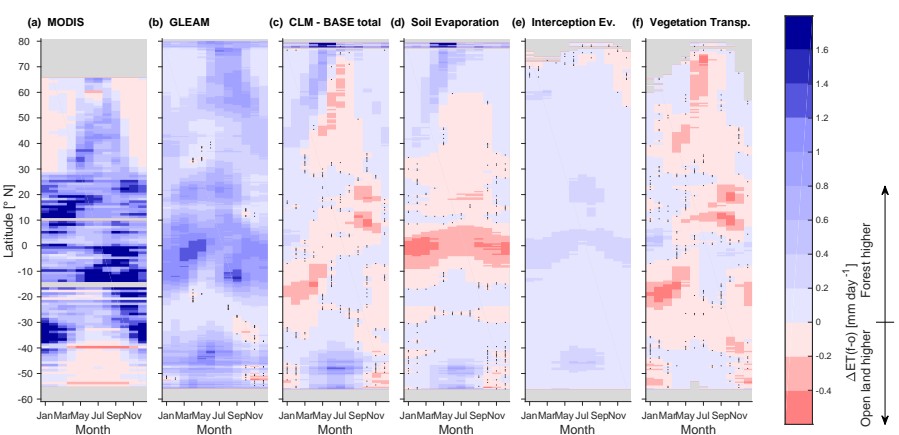

**Figure 4.** Seasonal and latitudinal variations of ΔET(f-o) in (a) MODIS, (b) GLEAM, and difference of forest minus open land in (c) total ET, (d) soil evaporation, (e) canopy interception evaporation, and (f) vegetation transpiration in CLM - BASE.



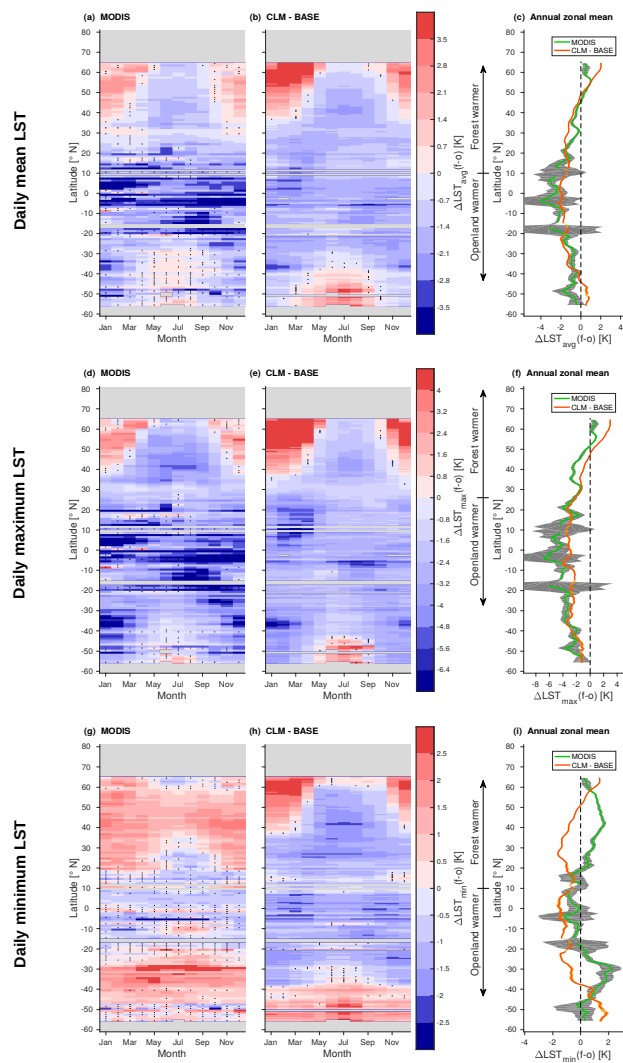

**Figure 5.** Seasonal and latitudinal variations of $\Delta LST_{avg}$(f-o) in (a) the MODIS observations and (b) CLM-BASE. Points with a mean which is insignificantly different from zero in a two-sided t-test at 95 % confidence level are marked with a black dot. Panel (c) shows the zonal annual mean of both MODIS (in green along with its confidence interval in grey) and CLM-BASE (in red, confidence interval in orange). Note that on this subfigure results have been smoothed with a 4° latitudinally-running mean. Only grid cells containing valid data in the MODIS observations were considered for the analysis of CLM-BASE. The same for $\Delta LST_{max}$(f-o) in panels (d),(e),(f) and for $\Delta LST_{min}$(f-o) in panels (g),(h),(i).





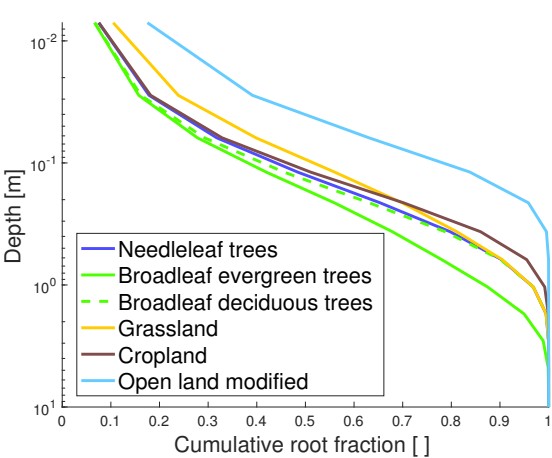

**Figure 6.** Cumulative root fraction (integrated from soil surface) of the different PFTs in the default version of CLM4.5 and in light blue the modified root fraction of open land PFTs used in CLM - PLUS. Note that the y-axis is logarithmic.



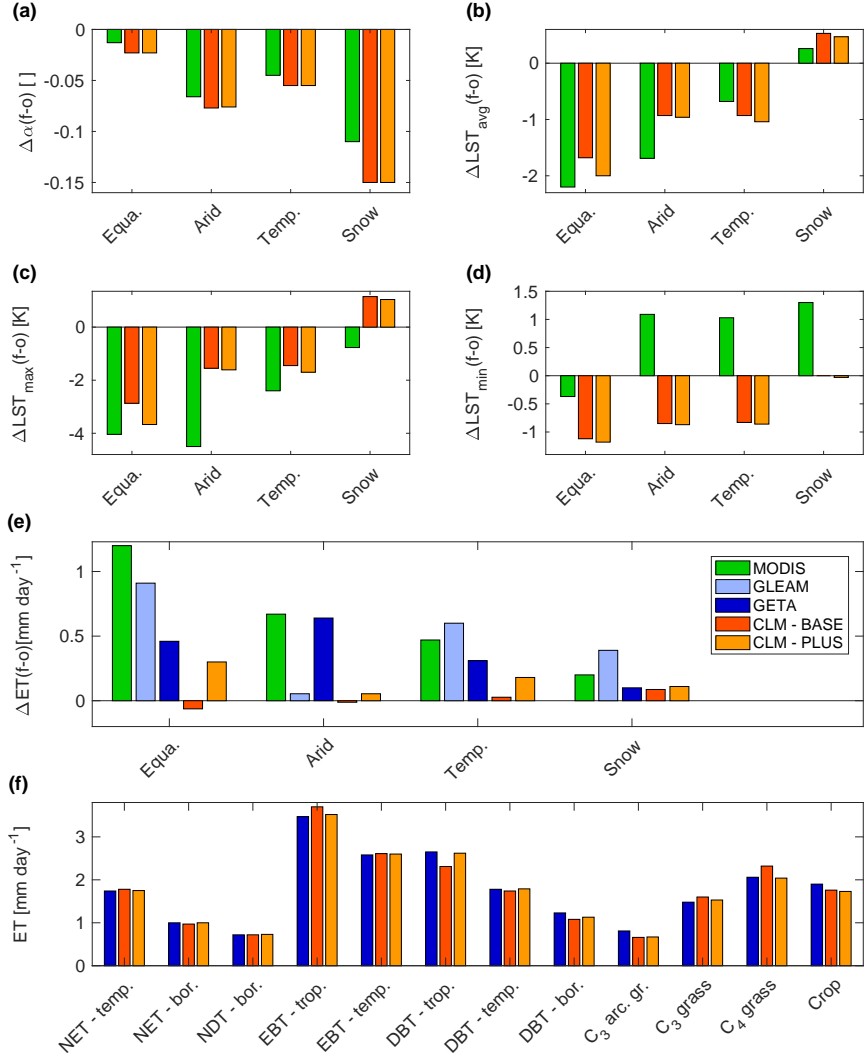

**Figure 7. Improvements in CLM - PLUS compared to CLM - BASE.** Area-weighted annual mean over Köppen-Geiger climate zones (Kottek et al., 2006) of (a) $\Delta\alpha$(f-o), (b) $\Delta LST_{avg}$(f-o), (c) $\Delta LST_{max}$(f-o), and (d) $\Delta LST_{min}$(f-o) in MODIS (green), CLM - BASE (red), and CLM - PLUS (orange). Only grid cells containing valid data in the MODIS observations were considered for analysis of CLM4.5. Panel (e) shows the area weighted mean over the Köppen-Geiger climate zone of $\Delta ET$(f-o) in MODIS (green), GLEAM (light blue), GETA (dark blue), CLM - BASE (red), and CLM - PLUS (orange) and panel (f) the area weighted mean ET for each PFT analyzed in this study according to the GETA (dark blue), CLM - BASE (red), and CLM - PLUS (orange). The acronyms of the PFTs are defined in Table 3.





**Table 1.** Overview of the different modifications of CLM4.5 incorporated in the simulations presented this study.

| Simulation | SeSCs | Shallow roots | 10 % | Light limitation | $V_{cmax}$ |
|---|---|---|---|---|---|
| CLM - DFLT | – | – | – | – | – |
| CLM - BASE | ✓ | – | – | – | – |
| CLM - PLUS | ✓ | ✓ | ✓ | ✓ | ✓ |





**Table 2.** Area-weighted annual mean of $\Delta\alpha$(f-o), $\Delta LST_{avg}$(f-o), $\Delta LST_{max}$(f-o), and $\Delta LST_{min}$(f-o) over the Köppen-Geiger climate zones (Kottek et al., 2006) in MODIS observations and CLM‑BASE. For these variables only grid cells containing valid data in the MODIS observations were considered for the analysis of CLM‑BASE. Area-weighted annual mean of $\Delta ET$(f-o) over the Köppen-Geiger climate zones in MODIS, GLEAM, GETA and CLM‑BASE. Also shown is the fraction of these climate zones covered by the MODIS $\Delta\alpha$(f-o) observations (note that these fractions vary slightly for different variables).

| Climate zone | Frac. [%] | $\Delta\alpha$(f-o) [ ] | | $\Delta ET$(f-o) [mm day$^{-1}$] | | | | $\Delta LST_{avg}$(f-o) [K] | | $\Delta LST_{max}$(f-o) [K] | | $\Delta LST_{min}$(f-o) [K] | |
|---|---|---|---|---|---|---|---|---|---|---|---|---|---|
| | | MODIS | CLM‑BASE | MODIS | GLEAM | GETA | CLM‑BASE | MODIS | CLM‑BASE | MODIS | CLM‑BASE | MODIS | CLM‑BASE |
| Equatorial | 6.6 | -0.013 | -0.023 | 1.2 | 0.92 | 0.46 | -0.063 | -2.2 | -1.7 | -4.0 | -2.9 | -0.37 | -1.1 |
| Arid | 1.7 | -0.066 | -0.077 | 0.67 | 0.048 | 0.64 | -0.012 | -1.7 | -0.93 | -4.5 | -1.6 | 1.1 | -0.85 |
| Warm temperate | 32 | -0.045 | -0.055 | 0.47 | 0.60 | 0.31 | 0.027 | -0.68 | -0.93 | -2.4 | -1.5 | 1.0 | -0.83 |
| Snow | 21 | -0.11 | -0.15 | 0.20 | 0.40 | 0.10 | 0.087 | 0.26 | 0.53 | -0.77 | 1.2 | 1.3 | 0.00 |





**Table 3.** Area-weighted annual mean ET for each PFT analyzed in this study according to the GETA data and in CLM - BASE as well as the fraction of the land surface covered by the different PFTs.

| Abbreviation | Full name | Frac. [%] | ET [mm day$^{-1}$] | |
| --- | --- | --- | --- | --- |
| | | | GETA | CLM - BASE |
| NET - temperate | Temperate evergreen needleleaf tree | 3.2 | 1.74 | 1.78 |
| NET - boreal | Boreal evergreen needleleaf tree | 6.9 | 1.00 | 0.97 |
| NDT - boreal | Boreal deciduous needleleaf tree | 1.0 | 0.72 | 0.72 |
| EBT - tropical | Tropical evergreen broadleaf tree | 9.5 | 3.47 | 3.70 |
| EBT - temperate | Temperate evergreen broadleaf tree | 1.5 | 2.58 | 2.61 |
| DBT - tropical | Tropical deciduous broadleaf tree | 8.0 | 2.65 | 2.31 |
| DBT - temperate | Temperate deciduous broadleaf tree | 3.1 | 1.78 | 1.74 |
| DBT - boreal | Boreal deciduous broadleaf tree | 1.3 | 1.23 | 1.08 |
| $C_3$ arctic grass | | 3.1 | 0.81 | 0.66 |
| $C_3$ grass | | 8.8 | 1.48 | 1.60 |
| $C_4$ grass | | 8.0 | 2.06 | 2.32 |
| Crop | Unmanaged rainfed $C_3$ crop | 10 | 1.90 | 1.76 |





**Table 4.** ET and latent heat flux in-situ observations from various studies and the values in CLM - BASE and CLM - PLUS at the respective locations.

| Study | Region | PFTs | Unit | Season | Obs. | CLM - BASE | CLM - PLUS |
|---|---|---|---|---|---|---|---|
| Da Rocha et al. (2004) | Amazon | EBT | mm day$^{-1}$ | Dry | 3.96 | 3.49 | 3.48 |
| | | | | Wet | 3.18 | 3.57 | 3.37 |
| | | | | All | 3.51 | 3.54 | 3.40 |
| Von Randow et al. (2004) | Amazon | EBT | W m$^{-2}$ | Dry | 108.6 | 82.9 | 90.8 |
| | | | | Wet | 104.5 | 113.9 | 108.9 |
| | | Grass | | Dry | 63.9 | 81.2 | 64.7 |
| | | | | Wet | 83.0 | 113.9 | 100.1 |
| Liu et al. (2005) | Alaska | Grass | W m$^{-2}$ | All | 16.1 | 16.4 | 16.8 |
| | | DBT | | All | 22.5 | 13.7 | 14.1 |
| | | ENF | | All | 23.9 | 18.0 | 18.4 |





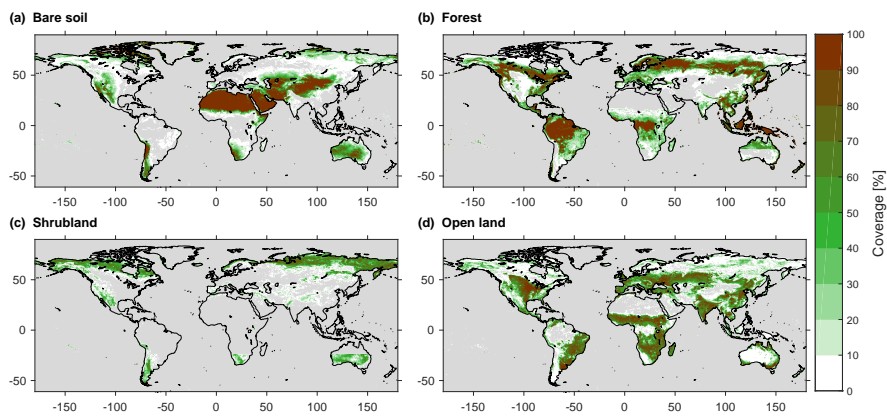

**Figure A1.** The fraction of the CLM4.5 grid cells covered by (a) bare soil, (b) forest, (c) shrubland, and (d) open land.





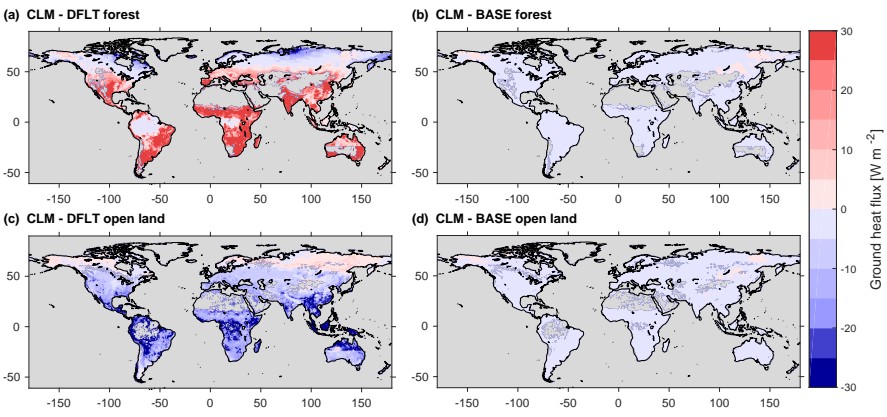

**Figure A2.** GH flux for forests (a and b) and open land (c and d) in CLM - DFLT (a and c) and CLM - BASE (b and d). Positive values correspond to a heat flux out of the soil.



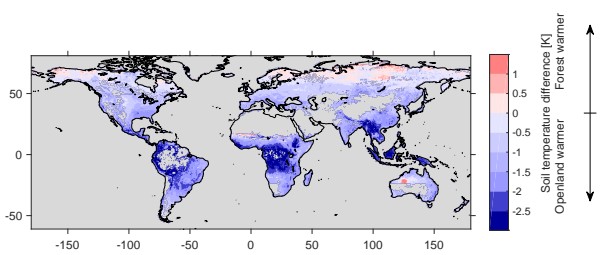

**Figure A3.** Difference in vertically-averaged annual mean soil temperature of forest minus open land in CLM - BASE.





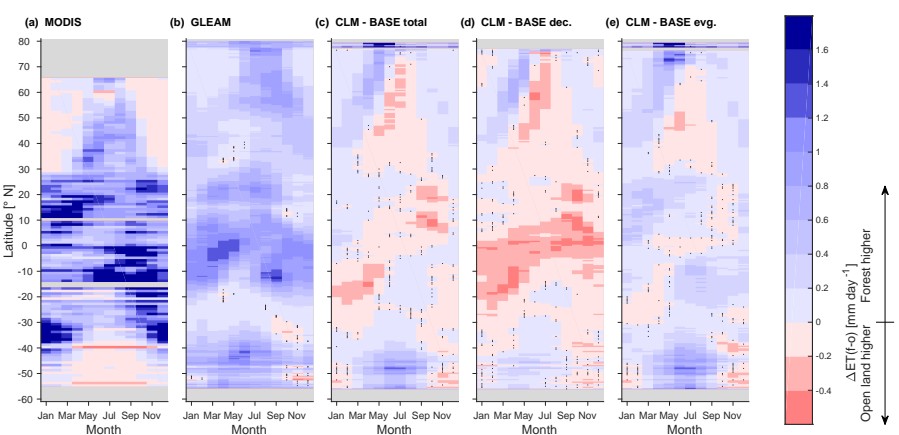

**Figure A4.** Seasonal and latitudinal variations of $\Delta$ET(f-o) in (a) MODIS, (b) GLEAM and, in CLM - BASE for (c) all tree PFTs minus open land, (d) deciduous tree PFTs only minus open land, (e) evergreen tree PFTs only minus open land.




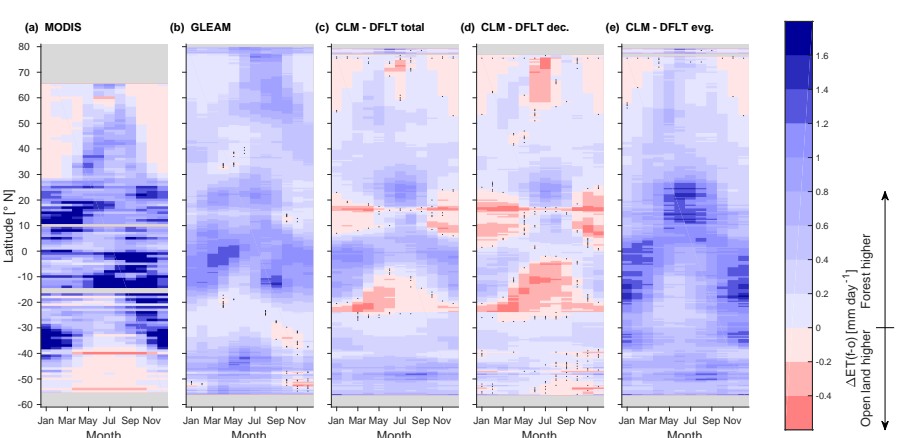

**Figure A5.** As Fig A4 but for CLM - DFLT.



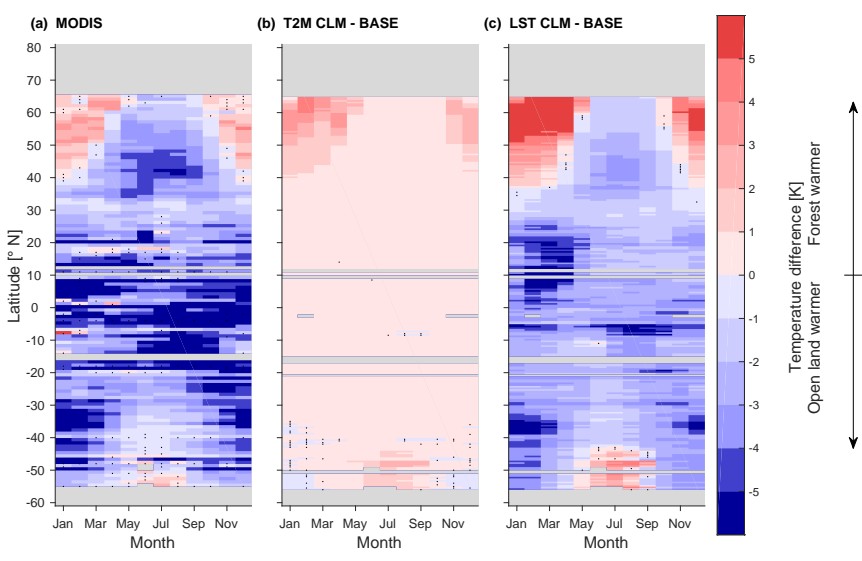

**Figure A6.** Seasonal and latitudinal variations of $\Delta\text{LST}_{max}$(f-o) in (a) the MODIS observations and in (c) CLM - BASE. Panel (b) shows the difference in daily maximum T2M in CLM - BASE.



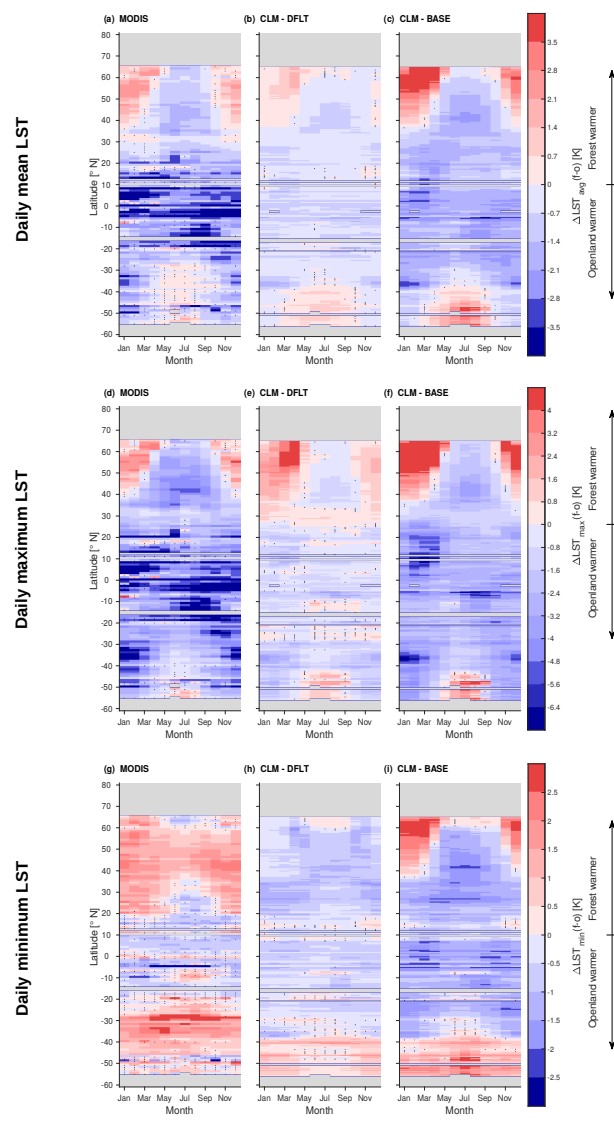

**Figure A7.** Seasonal and latitudinal variations of $\Delta LST_{avg}$(f-o) in (a) the MODIS observations, (b) CLM - DFLT, and (c) CLM - BASE. Points with a mean which is insignificantly different from zero in a two-sided t-test at 95 % confidence level are marked with a black dot. Only grid cells containing valid data in the MODIS observations were considered for the analysis of CLM - DFLT and CLM - BASE. The same for $\Delta LST_{max}$(f-o) in panels (d),(e),(f) and for $\Delta LST_{min}$(f-o) in panels (g),(h),(i).



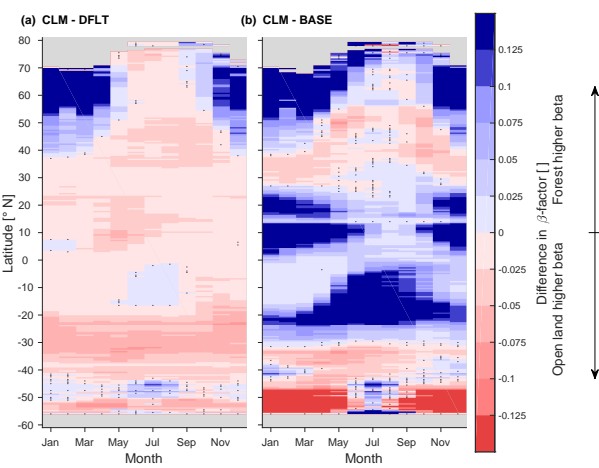

**Figure A8.** Seasonal and latitudinal variations of $\beta_t$-factor differences of forest minus open land in (a) CLM - DFLT and (b) CLM - BASE.



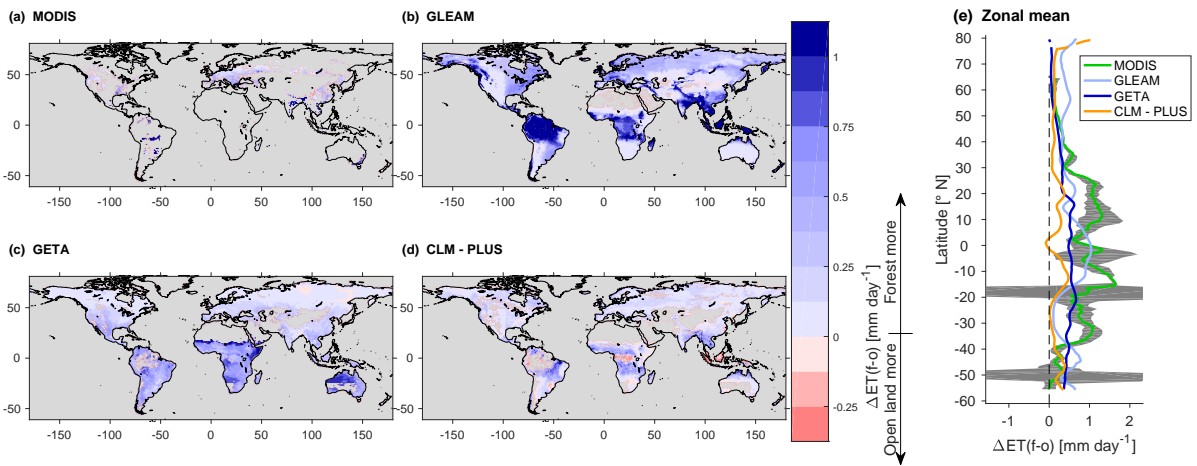

**Figure A9.** As Fig. 2 but for the CLM - PLUS.





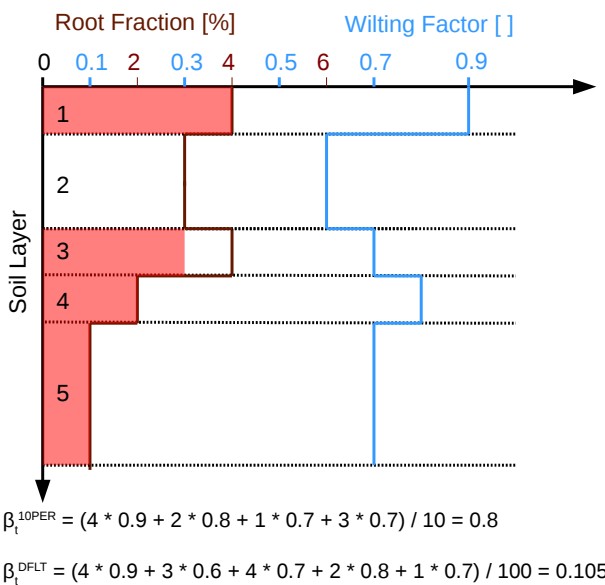

$\beta_t^{10PER}$ = (4 * 0.9 + 2 * 0.8 + 1 * 0.7 + 3 * 0.7) / 10 = 0.8

$\beta_t^{DFLT}$ = (4 * 0.9 + 3 * 0.6 + 4 * 0.7 + 2 * 0.8 + 1 * 0.7) / 100 = 0.105

**Figure A10. Example of the calculation of the $\beta_t$-factor with the 10 % modification.** Shown are five soil layers with the fraction of the roots in these layers in brown and the wilting factor in blue. On the bottom the calculation of $\beta_t$ for this particular example with the 10 % modification ($\beta_t^{10PER}$) and the default calculation in CLM4.5 ($\beta_t^{DFLT}$), assuming the roots not shown have a wilting factor of zero. The root fractions eventually used to calculate $\beta_t^{10PER}$ are shaded in red.





**Table A1.** The default PFT classification in CLM4.5.

| No. | Abbr. | Full name | Class |
|---|---|---|---|
| 1 | Bare soil | | – |
| 2 | NET - temperate | Temperate evergreen needleleaf tree | Tree |
| 3 | NET - boreal | Boreal evergreen needleleaf tree | Tree |
| 4 | NDT - boreal | Boreal deciduous needleleaf tree | Tree |
| 5 | BET - tropical | Tropical evergreen broadleaf tree | Tree |
| 6 | BET - temperate | Temperate evergreen broadleaf tree | Tree |
| 7 | BDT - tropical | Tropical deciduous broadleaf tree | Tree |
| 8 | BDT - temperate | Temperate deciduous broadleaf tree | Tree |
| 9 | BDT - boreal | Boreal deciduous broadleaf tree | Tree |
| 10 | BES - temperate | Temperate evergreen broadleaf shrub | – |
| 11 | BDS - temperate | Temperate deciduous broadleaf shrub | – |
| 12 | BDS - boreal | Boreal deciduous broadleaf shrub | – |
| 13 | $C_3$ arctic grass | | Open land |
| 14 | $C_3$ grass | | Open land |
| 15 | $C_4$ grass | | Open land |
| 16 | Crop | Unmanaged rainfed $C_3$ crop | Open land |

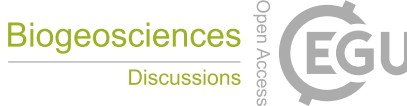



**Table A2.** The land cover types from Ambrose and Sterling (2014) (GETA) used in this study and the number of respective PFTs in CLM4.5 applied to the different land cover types (Table A1).

| Abbr. GETA | Full name GETA | PFTs of CLM4.5 |
|---|---|---|
| ENF | Evergreen needleleaf forest | 2, 3 |
| DNF | Deciduous needleleaf forest | 4 |
| EBF | Evergreen broadleaf forest | 5, 6 |
| DBF | Deciduous broadleaf forest | 7, 8, 9 |
| GRS | Grassland | 13, 14, 15 |
| CRN | Non-irrigated cropland | 16 |



**Table A3.** The PFT-specific values of $V_{cmax}$ [µmol m$^{-2}$ s$^{-1}$], $r_a$, and $r_b$ [ ] in default of CLM4.5 and in CLM - PLUS.

| PFT name | Default | | | CLM - PLUS | | |
|---|---|---|---|---|---|---|
| | $r_a$ | $r_b$ | $V_{cmax}$ | $r_a$ | $r_b$ | $V_{cmax}$ |
| NET - temperate | 7.0 | 2.0 | 62.5 | default | | 50 |
| NET - boreal | 7.0 | 2.0 | 62.6 | default | | default |
| NDT - boreal | 7.0 | 2.0 | 39.1 | default | | default |
| EBT - tropical | 7.0 | 1.0 | 55.0 | default | | 35 |
| EBT - temperate | 7.0 | 1.0 | 61.5 | default | | 50 |
| DBT - tropical | 6.0 | 2.0 | 41.0 | default | | 65 |
| DBT - temperate | 6.0 | 2.0 | 57.7 | default | | default |
| DBT - boreal | 6.0 | 2.0 | 57.7 | default | | 70 |
| $C_3$ arctic grass | 11.0 | 2.0 | 78.2 | 11.0 | 11.0 | 90 |
| $C_3$ grass | 11.0 | 2.0 | 78.2 | 11.0 | 11.0 | 60 |
| $C_4$ grass | 11.0 | 2.0 | 51.6 | 11.0 | 11.0 | default |
| Crop | 6.0 | 3.0 | 100.7 | 11.0 | 11.0 | 90 |




**Table A4.** Overview of the different modifications of CLM4.5 incorporated in the simulations presented this study.

| Run | SeSCs | Shallow roots | 10 % | Light limitation | $V_{cmax}$ |
|---|---|---|---|---|---|
| CLM - DFLT | – | – | – | – | – |
| CLM - BASE | ✓ | – | – | – | – |
| CLM - ROOT | ✓ | ✓ | – | – | – |
| CLM - 10PER | ✓ | ✓ | ✓ | – | – |
| CLM - LIGHT | ✓ | ✓ | ✓ | ✓ | – |
| CLM - PLUS | ✓ | ✓ | ✓ | ✓ | ✓ |





**Table A5.** Area-weighted annual mean ET for each PFT analyzed in this study according to the GETA data and in the different configurations of CLM4.5 and fraction of the land surface covered by the different PFTs. On the bottom is listed the global integral of annual ET.

| Abbr. | Full name | Frac. [%] | ET [mm day$^{-1}$] | | | | | |
|---|---|---|---|---|---|---|---|---|
| | | | GETA | BASE | ROOT | 10PER | LIGHT | PLUS |
| NET - temperate | Needleleaf evergreen tree - temperate | 3.2 | 1.74 | 1.78 | 1.78 | 1.81 | 1.84 | 1.75 |
| NET - boreal | Needleleaf evergreen tree - boreal | 6.9 | 1.00 | 0.97 | 0.97 | 0.98 | 1.00 | 1.00 |
| NDT - boreal | Needleleaf deciduous tree - boreal | 1.0 | 0.72 | 0.72 | 0.72 | 0.72 | 0.73 | 0.73 |
| EBT - tropical | Broadleaf evergreen tree - tropical | 9.5 | 3.47 | 3.70 | 3.70 | 3.78 | 3.87 | 3.52 |
| EBT - temperate | Broadleaf evergreen tree - temperate | 1.5 | 2.58 | 2.61 | 2.61 | 2.66 | 2.70 | 2.60 |
| DBT - tropical | Broadleaf deciduous tree - tropical | 8.0 | 2.65 | 2.31 | 2.31 | 2.42 | 2.44 | 2.62 |
| DBT - temperate | Broadleaf deciduous tree - temperate | 3.1 | 1.78 | 1.74 | 1.74 | 1.76 | 1.79 | 1.79 |
| DBT - boreal | Broadleaf deciduous tree - boreal | 1.3 | 1.23 | 1.08 | 1.08 | 1.08 | 1.10 | 1.13 |
| $C_3$ arctic grass | | 3.1 | 0.81 | 0.66 | 0.65 | 0.65 | 0.66 | 0.67 |
| $C_3$ grass | | 8.8 | 1.48 | 1.60 | 1.53 | 1.56 | 1.57 | 1.53 |
| $C_4$ grass | | 8.0 | 2.06 | 2.32 | 2.18 | 2.22 | 2.12 | 2.04 |
| Crop | $C_3$ unmanaged rainfed crop | 10 | 1.90 | 1.76 | 1.70 | 1.73 | 1.74 | 1.73 |
| Total ET [km$^3$ yr$^{-1}$] | | | | 70223 | 69059 | 70322 | 70649 | 69023 |



**Table A6.** ET and latent heat flux in-situ observations from various studies and the values of the different CLM4.5 sensitivity tests at the respective locations.

| Study | Region | PFTs | Unit | Season | Obs. | BASE | ROOT | 10PER | LIGHT | PLUS |
|-------|--------|------|------|--------|------|------|------|-------|-------|------|
| Da Rocha et al. (2004) | Amazon | EBT | mm day$^{-1}$ | Dry | 3.96 | 3.49 | 3.49 | 3.90 | 4.06 | 3.48 |
| | | | | Wet | 3.18 | 3.57 | 3.57 | 3.57 | 3.64 | 3.37 |
| | | | | All | 3.51 | 3.54 | 3.54 | 3.68 | 3.79 | 3.40 |
| Von Randow et al. (2004) | Amazon | EBT | W m$^{-2}$ | Dry | 108.6 | 82.9 | 82.9 | 100.6 | 105.3 | 90.8 |
| | | | | Wet | 104.5 | 113.9 | 113.9 | 113.8 | 116.2 | 108.9 |
| | | Grass | | Dry | 63.9 | 81.2 | 56.0 | 60.2 | 62.7 | 64.7 |
| | | | | Wet | 83.0 | 113.9 | 113.9 | 113.9 | 106.1 | 100.1 |
| Liu et al. (2005) | Alaska | Grass | W m$^{-2}$ | All | 16.1 | 16.4 | 16.8 | 16.8 | 16.8 | 16.8 |
| | | DBT | | All | 22.5 | 13.7 | 13.7 | 14.0 | 14.0 | 14.1 |
| | | ENF | | All | 23.9 | 18.0 | 18.0 | 18.4 | 18.4 | 18.4 |