# Peer review of "Evaluating and Improving the Community Land Model's Sensitivity to Land Cover"

_Biogeosciences, 2017_

## Referee Comment (RC1) · Anonymous Referee #1 · 7 Feb 2018

Well written manuscript. The subject is relevant and timely. I would recommend publication in Biogeosciences after the following revisions are made.

From the stylistic point of view, in my opinion the paper would read better if it is moderately restructured. The description of the sensitivity experiment (section 3.2) should be located in the methodology section before the result. This would avoid the feeling of jumping back and forth from results to methods, and would help justify the presence of CLM-PLUS results in the earlier figures (before its description). I know this may feel awkward as the CLM – PLUS simulation is seen as a response to the problems (i.e. results) identified in section 3.1, but with some effort I am confident the restructuring can be done. I would introduce this idea (that a CLM – PLUS simulation is done as a response to the first results) in the last paragraph of the introduction, and then describe

it in the last section of the methods, saying you are anticipating (in the text) the results that will be presented thereafter.

Another restructuring point I would strongly recommend is to try to separate Results from Discussion. The combined section currently works quite well for 'Results', as such a section should not be just a description of results but also an interpretation of them. But some parts can be moved to a more general 'discussion' section in which the whole approach is discussed in a broader sense, providing more insight of the caveats and advantages of the whole experiment, and how it relates to the broader picture in Earth System Science.

Deforestation is more complex than a simple transition from forest to open land described in the Li et al 2015 MODIS dataset, as different types of forest (e.g. evergreen or deciduous) would have different effects (on snow masking and albedo for instance), and different kind of open lands will also behave differently (management would arguably have a strong influence). With the GETA data, the authors do explore this variability for ET to some extent. In my opinion a more thorough discussion is warranted, even if further analyses are not required within this study. Could anything be said on PFT specific differences for albedo and LST? Are there other field-based datasets such as GETA that could be used for these variables? Could other datasets from remote sensing that differentiate amongst forest types be used? If not, mentioning this need could justify and stimulate the development of such products in the future.

There are some doubts on how comparable the deltas that are extracted from GLEAM are with respect to the Li MODIS dataset and to the CLM sub-grid simulations. If I understand correctly, GLEAM provides separate values for tall canopies and low vegetation over the same 0.25 dd pixels, and to obtain a change between 'forest' to 'openland', one makes the difference a pixel level between the value for tall canopies and for low vegetation. However, to understand better the possible repercussions this may have on the analysis, it would be necessary to have more information on how the distinction between tall canopies and low vegetation is made in GLEAM. What land cover

maps are used (if any)? How do these match with the CLM distribution of PFTs?

The MODIS Li et al. 2015 dataset depends on setting a threshold on the percentage of forest/trees that there are in a pixel so as to consider it 'forest' or 'openland'. They also show in their supplementary material that the choice of the threshold does have some effect of the results. How does this affect the comparison with CLM sub-grid results, for which the signal is fully 'un-mixed'? In my understanding this has the effect that the MODIS delta will often relate to a comparison from a 'not-so-full-forest' to a 'not-so-treeless-openland', while the simulations are from a 'full-closed forest' to a 'treeless openland'. How does this impact the results? Can something be done about it? Regarding the discussion on T2M vs LST in both the models and observations, an importantpoint that is not completely clear is whether T2M is considered as 2m above the canopy or above the soil (i.e. within the forest). Note that in studies like Alkama & Cescatti (2016), the techniques to obtain T2M from satellite LST require weather stations, which typically use WMO definitions by which temperature is measured above a standard grass canopy, even if it is surrounded by forest. This means that the T2M obtained is not that which is observed within the canopy (i.e. under the trees) nor the one above the trees. In the model, and hence in this analysis, what temperature are we speaking about and how can the comparability between observations and models be ensured?

Other punctual remarks include:

Page 6, Line 3: Style: avoid starting sentence with a number

Page 9, Line 21: You need to specify that you are speaking of the MODIS instrument on-board of the the AQUA satellite. There is also a MODIS instrument on-board of the TERRA satellite for which the overpass time (at the Equator) is approx. 10:30 am and 10:30 pm. The choice of AQUA is sound as those times are actually not that far from the maximum and minimum

Page 9 Line 31, Could you speculate on why the model would have this behaviour?

Page 11, Line 19... Could you add some info on whether this shallower root distribution is closer to what is observed, perhaps based on information from the references cited (Fan2017 & Canadell1996)? Ideally it would be good to have a line in Fig 6 for the observations over 'openland'.

Fig 1, 2, How do you calculate the confidence interval in MODIS? Do they come from the original product of Li et al. 2015? If so, do explain a bit more how they are calculated and how should the reader interpret it?

For all plots like that of figure 1, I am not too sure how much we gain in insight by having the fine 1dd resolution. I would recommend using broader latitudinal bins (e.g. 2.5 or perhaps even 5 dd) so as to have larger boxes in which the points of the t-test are larger and clearer.

---

## Referee Comment (RC2) · Anonymous Referee #2 · 24 Mar 2018

Review of bg-2017-501: Evaluating and improving the Community Land Model's sensitivity to land cover

Summary

The authors compare CLM modeled differences in biophysical properties between forest and open land with remote sensing estimates (MODIS-based) and additional observational data for evapotranspiration (ET). They find that albedo and average and max surface radiative temperature differences are adequately simulated by CLM, but min temperature and ET are not. They note that CLM with separate pft soil columns performs better than the default shared soil column, except for ET. They conclude that error in forest transpiration parameters/processes is responsible for the relatively poor ET performance of CLM, and make modifications to improve the ET response. These

modifications improve the ET response somewhat, and mostly in lower latitudes, but discrepancies with observations still remain.

Overall the paper is clearly written and the methods are sound. It advances our understanding of land change effects on surface properties and how to go about evaluating and estimating such effects. My main concerns are with some of the interpretations of the results, and the premature conclusion that vegetation transpiration is the main source of ET errors in the model. To make this conclusion the authors need to carry out further tests related to soil evaporation and potentially interception evaporation as well (as this may be compensating to some degree for error in soil evaporation). These are the main issues that need to be addressed for publication (more detail is provided below):

1) Complete the analysis of the sources of error. You test only things related to vegetation transpiration and not soil evaporation. Your data do not clearly indicate that vegetation is the main driver, and in fact show that soil evaporation could also be a dominant source of error. Just because your modifications for transpiration show some improvement does not mean that they are correct, because you could be over-compensating or over-fitting these parameter values.

2) Please provide a metric for quantifying the effects of the modifications. Figure 7 (and the aggregate climate zones) is not adequate for demonstrating significant improvement of the results due to the modifications.

Specific comments and suggestions:

Abstract

Introduction

Methods and data

page 6 lines 7-11: I think CLM also outputs a surface radiative temperature. Why didn't you use this?

Results and discussion

page 7 line 21: confidence in which observations? the non-outliers i assume.

page 7 line 28: be clear that it is the deciduous/evergreen trees in the model that are the source

page 8 lines 21-22 and 31-32: this statement is not supported by your data or the rest of this paragraph. while the visual pattern between the VTR and total is similar, the soil evap effects are compensated for by the interception effects, thus leaving VTR to dominate the pattern. but this doesn't mean that the soil evap is not a main contributor, especially in the tropics. and you mention the biases in the non-forest that contribute to this discrepancy as well. Figure 4 also indicates that the soil evaporation dominates the total ET pattern in the higher latitudes, which is where your modifications show little improvement.

page 9 line 8 should be referencing table 2 here

page 9 line 25: at all latitudes

page 9 line 26: same sign as delta LST

page 10 lines 5-6: if comparing for lee et al, why reference alkama and cescatti for the amplification? you should include the delta LST per degree from lee et al for a consistent comparison, and to show that these observations also show this amplification

page 11 line 5: not sure that this is the case

page 12 lines 11-22: this indicates that your hypothesis regarding VTR as the main driver of discrepancies may not be correct. while you get improvements, soil evap remains a problem, and you may even be overcompensating with the VTR related modifications

Also, while the pft level comparison with GETA looks good, the climate zone comparison is more difficult to evaluate. Aggregating to these climate zones smooths out a

lot of spatial variability, and may be too coarse to adequately evaluate the modifications. can you calculate a metric to quantify the effects of the modifications? what do pixel-level correlations between the model and the obs look like? are these correlations improved by the modifications? would zonal grouping make more sense than the climate zones?

page 13 line 18: is this because you used prescribed atmospheric forcing?

page 13 lines 29-30 i am still not completely convinced of this

page 15 lines 7-8: this suggests that soil evaporation may also be a main factor

Figures and tables

Generally, why show a CI for only the modis zonal average? What about the other data and the model outputs? And is CI the best metric to depict variability here? There are many reasons for variability around the globe at a given latitude (e.g., different weather patterns, continental vs maritime), and we should not expect a zonal mean to behave like a population mean estimate that supposedly characterizes a more homogenous group.

Figure 2a: this does not appear to be the correct figure. it does not match with the averages in panel 2c, nor table 2

Figure 5 row labels do not that these are differences, which can be confusing

---

## Author Comment (AC4) · 24 Apr 2018

We sincerely thank the referee for her/his very constructive review and the important points raised. We think that we can improve the quality of our manuscript substantially due to the inputs of the referee. Hereafter the referee's comments are reported in black and our replies are highlighted in blue.

*From the stylistic point of view, in my opinion the paper would read better if it is moderately restructured. The description of the sensitivity experiment (section 3.2) should be located in the methodology section before the result. This would avoid the feeling of jumping back and forth from results to methods, and would help justify the presence of CLM-PLUS results in the earlier figures (before its description). I know this may feel awkward as the CLM – PLUS simulation is seen as a response to the problems (i.e. results) identified in section 3.1, but with some effort I am confident the restructuring can be done. I would introduce this idea (that a CLM – PLUS simulation is done as a response to the first results) in the last paragraph of the introduction, and then describe it in the last section of the methods, saying you are anticipating (in the text) the results that will be presented thereafter.*

**Answer:** We agree that the experiment description in section 3.2 could be placed elsewhere. We decided to add the methodological details to the appendix dedicated to the sensitivity experiments including the detailed description of the implementation. We will focus only on the results in section 3.2 and give a brief overview of the sensitivity experiments in the method section with a reference to the appendix for the more complete details. As the referee proposes we will describe the overall goal/motivation of the sensitivity experiment in the last part of the introduction.

*Another restructuring point I would strongly recommend is to try to separate Results from Discussion. The combined section currently works quite well for 'Results', as such a section should not be just a description of results but also an interpretation of them. But some parts can be moved to a more general 'discussion' section in which the whole approach is discussed in a broader sense, providing more insight of the caveats and advantages of the whole experiment, and how it relates to the broader picture in Earth System Science*

**Answer:** This is a good suggestion. We will add a separate Discussion section in the revised manuscript. Many of the points made below by the referees will effectively provide relevant material for the discussion section.

*Deforestation is more complex than a simple transition from forest to open land described in the Li et al 2015 MODIS dataset, as different types of forest (e.g. evergreen or deciduous) would have different effects (on snow masking and albedo for instance), and different kind of open lands will also behave differently (management would arguably have a strong influence). With the GETA data, the authors do explore this variability for ET to some extent. In my opinion a more thorough discussion is warranted, even if further analyses are not required within this study. Could anything be said on PFT specific differences for albedo and LST? Are there other field-based datasets such as GETA that could be used for these variables? Could other datasets from remote sensing that differentiate amongst forest types be used? If not, mentioning this need could justify and stimulate the development of such products in the future.*

**Answer:** We agree with this point and we will introduce a more complete discussion of this topic in the new discussion section (we already touched upon the issue by mentioning the need to distinguish between irrigated and rainfed crops in p. 7 ll. 21-24). In particular, we will add a reference to the new remote sensing-based dataset by Duveiller et al., (2018) which was released only after we submitted our manuscript. This dataset - distinguishing between different forest and open land types - is a promising element towards refining the type of evaluation strategy we presented here.

*There are some doubts on how comparable the deltas that are extracted from GLEAM are with respect to the Li MODIS dataset and to the CLM sub-grid simulations. If I understand correctly, GLEAM provides separate values for tall canopies and low vegetation over the same 0.25 dd pixels, and to obtain a change between 'forest' to 'open-land', one makes the difference a pixel level between the value for tall canopies and for low vegetation. However, to understand better the possible repercussions this may have on the analysis, it would be necessary to have more information on how the distinction between tall canopies and low vegetation is made in GLEAM. What land cover maps are used (if any)? How do these match with the CLM distribution of PFTs?*

**Answer:** This is an important point which we will discuss in more detail in the revised manuscript. Indeed, GLEAM and Li et al. MODIS data are not based on the same land cover information (GLEAM uses MOD44B whereas Li et al. (2015) use MCD12C1). While MOD44B provides the fraction of a grid cell covered by trees, non-tree vegetation, and non-vegetated land surfaces, MCD12C1 provides the dominant IGBP land cover type in each pixel. The non-tree vegetation fraction of MOD44B incorporates shrubland which is excluded in MODIS and our CLM analysis. This might be the cause for the much lower value of the mean ET difference in the arid climate zone for GLEAM compared to the other two data sets (MODIS and GETA).

*The MODIS Li et al. 2015 dataset depends on setting a threshold on the percentage of forest/trees that there are in a pixel so as to consider it 'forest' or 'openland'. They also show in their supplementary material that the choice of the threshold does have some effect of the results. How does this affect the comparison with CLM sub-grid results, for which the signal is fully 'un-mixed'? In my understanding this has the effect that the MODIS delta will often relate to a comparison from a 'not-so-full-forest' to a 'not-so-treeless-openland', while the simulations are from a 'full-closed forest' to a 'treeless openland'. How does this impact the results? Can something be done about it?*

**Answer:** This reasoning is indeed correct. This issue was already mentioned in previous studies comparing satellite-derived albedo products with in situ measurements, as we discuss on page 6 lines 26-30 of the initial manuscript. However, this of course also applies to the other variables and we will therefore add this point in the discussion section.
Recently, Duveiller et al. (2018) established a relationship between vegetation cover fractions and different surface variables in similar moving windows as Li et al. (2015) to estimate the signal of vegetation cover changes to avoid this issue. As mentioned in our response to the 3rd comment, we encourage using this data set for future studies of a similar kind as ours and will make this clear in the revised manuscript.

*Regarding the discussion on T2M vs LST in both the models and observations, an*

*Important point that is not completely clear is whether T2M is considered as 2m above the canopy or above the soil (i.e. within the forest). Note that in studies like Alkama & Cescatti (2016), the techniques to obtain T2M from satellite LST require weather stations, which typically use WMO definitions by which temperature is measured above a standard grass canopy, even if it is surrounded by forest. This means that the T2M obtained is not that which is observed within the canopy (i.e. under the trees) nor the one above the trees. In the model, and hence in this analysis, what temperature are we speaking about and how can the comparability between observations and models be ensured?*

**Answer:** T2M in CLM4.5 is defined as the temperature 2 m above the apparent sink for sensible heat (Oleson et al., 2013; Eq. 5.58) which lies within the canopy air space. In the manuscript we argue that T2M is not the right temperature diagnostic to compare to LST observations (this is why we recalculated a radiative temperature (TRAD) based on the outgoing longwave radiation). We nevertheless show a comparison with daily maximum T2M difference in the appendix (Fig. A6) to highlight the different sign of the response in T2M compared to TRAD in CLM4.5. This result is surprising and is worth noting since it implies that modelling studies looking at land use effects might be affected by the choice of temperature diagnostics, which is an issue that has been overlooked in our community. That said, evaluating this T2M temperature signal in CLM4.5 against observations is very challenging since, as the referee rightfully points out, the WMO T2M concept is by definition not applicable to forest and therefore "T2M" in forest is ill-defined. For instance, the measurements of Lee et al. (2011) report "T2M" above the canopy and "T2M" in Alkama & Cescatti (2016) is indeed a mixed concept derived empirically, which is yet another definition compared to the CLM4.5 definition above. We will add these clarifications to the discussion. Also, we will replace Fig. A6 in the manuscript with the figure below to make it clearer for the reader that we do not intend to compare the T2M signal in CLM with the MODIS LST observations but emphasize that the T2M and the TRAD signal in CLM look very different.

[Figure]

**Revised Figure A6**. Seasonal and latitudinal variations of (a) the daily maximum T2M difference of forest minus open land and (b) LSTmax(f-o) in CLM- BASE.

*Page 9 Line 31, Could you speculate on why the model would have this behaviour?*

**Answer:** We did not further investigate why the model exhibits this behavior. We think that we can exclude the fact that our simulations were made in offline mode as the cause, since online simulations using CLM exhibit similar daily maximum T2M signals, as mentioned in the original manuscript (p. 8 ll. 28-31).

*Could you add some info on whether this shallower root distribution is closer to what is observed, perhaps based on information from the references cited (Fan2017 & Canadell1996)? Ideally it would be good to have a line in Fig 6 for the observations over 'openland'.*

**Answer:** We will visualize the rooting depths reported in Fan et al. (2017) in Fig. 6. Note that when creating this new root distribution, the aim was not to fit the new distribution to a particular root distribution found in the literature. It was rather a test on how the model would react to a shallower distribution for open land. In fact, the resulting root profile for open land PFTs is rather shallow compared to what is observed but still within the observational range. We will mention this in the discussion of the sensitivity experiment.

*How do you calculate the confidence interval in MODIS? Do they come from the original product of Li et al. 2015? If so, do explain a bit more how they are calculated and how should the reader interpret it?*

**Answer:** The confidence interval is the original one from Li et al. (2015). It is the two-sided 95% confidence interval estimated by a t-test. As the 2nd referee mentions, the data we plot are not normally distributed. We will thus replace the confidence intervals with the interquartile range which is more suitable to visualize the variability of not-normally-distributed data.

*For all plots like that of figure 1, I am not too sure how much we gain in insight by having the fine 1dd resolution. I would recommend using broader latitudinal bins (e.g. 2.5 or perhaps even 5 dd) so as to have larger boxes in which the points of the t-test are larger and clearer.*

**Answer:** At 0.5° resolution the points displaying the results of the t-test are indeed hard to see. We will therefore average the CLM and GLEAM data to latitudinal bands of 1° for those plots.

References:

Bright, R. M., Davin, E. L., O'Halloran, T., Pongratz, J., Zhao, K., and Cescatti, A.: Local temperature response to land cover and management change driven by non-radiative processes, Nat. Clim. Change, 7, 296–302, https://doi.org/10.1038/nclimate3250, 2017.

Duveiller, G., Forzieri, G., Robertson, E., Li, W., Georgievski, G., Lawrence, P., Wiltshire, A., Ciais, P., Pongratz, J., Sitch, S., Arneth, A., and Cescatti, A. (2018). Biophysics and

vegetation cover change: a process-based evaluation framework for confronting land surface models with satellite observations.Earth System Science Data Discussions, 2018:1-24.

Fan, Y., Miguez-Macho, G., Jobbágy, E. G., Jackson, R. B., and Otero-Casal, C.: Hydrologic regulation of plant rooting depth, P. Natl. Acad. 10 Sci. USA, 114, 10 572–10 577, https://doi.org/10.1073/pnas.1712381114, 2017.

Lee, X., Goulden, M. L., Hollinger, D. Y., Barr, A., Black, T. A., Bohrer, G., Bracho, R., Drake, B., Goldstein, A., Gu, L., Katul, G., Kolb, T., Law, B. E., Margolis, L. H., Meyers, T., Monson, R., Munger, W., Oren, R., Paw U, K. T., Richardson, A. D., Schmid, H. P. Staebler, R.,Wofsy, S., and Zhao, L.: Observed increase in local cooling effect of deforestation at higher latitude, Nature, 479, 384–387, https://doi.org/10.1038/nature10588, 2011.

Oleson, K. W., Lawrence, D., B. Bonan, G., Drewniak, B., Huang, M., Koven, C., Levis, S., Li, F., Riley, W., M. Subin, Z., C. Swenson, S., E. Thornton, P., Bozbiyik, A., Fisher, R., Heald, C., Kluzek, E., Lamarque, J.-F., Lawrence, P., Leung, L., and Yang, Z.-L.: Technical Description of version 4.5 fo the Community Land Model (CLM), https://doi.org/10.5065/D6RR1W7M, 2013.

---

## Author Comment (AC5) · 24 Apr 2018

We thank the referee for her/his very useful comments which will help improve the manuscript severely. It appears that the referee is highly experienced in this field of research. In the following, we list the referee's comments in black and our replies in blue. Note that the order of the comments was changed to facilitate our response to related comments.

*1) Complete the analysis of the sources of error. You test only things related to vegetation transpiration and not soil evaporation. Your data do not clearly indicate that vegetation is the main driver, and in fact show that soil evaporation could also be a dominant source of error. Just because your modifications for transpiration show some improvement does not mean that they are correct, because you could be over-compensating or over-fitting these parameter values.*

*page 8 lines 21-22 and 31-32: this statement is not supported by your data or the rest of this paragraph. while the visual pattern between the VTR and total is similar, the soil evap effects are compensated for by the interception effects, thus leaving VTR to dominate the pattern. but this doesn't mean that the soil evap is not a main contributor, especially in the tropics. and you mention the biases in the non-forest that contribute to this discrepancy as well. Figure 4 also indicates that the soil evaporation dominates the total ET pattern in the higher latitudes, which is where your modifications show little improvement.*

*page 12 lines 11-22: this indicates that your hypothesis regarding VTR as the main driver of discrepancies may not be correct. while you get improvements, soil evap remains a problem, and you may even be overcompensating with the VTR related modifications*

**Answer:** We agree that the soil and canopy evaporation are both important for the creation of the total ET signal. In the manuscript this is not clear enough and the importance of VTR is emphasized too much. Thus, we will rephrase some of the statements in the manuscript (e.g., page 8 lines 21-22 and 31-32 and page 12 lines 11-22).
We tried to assess the importance of the individual components more objectively by calculating the pearson correlation and the index of agreement (Duveiller et al., 2016) between the monthly difference of forest minus open land of these individual components and the monthly difference in total ET. The difference in vegetation transpiration in CLM4.5 exhibits a stronger correlation over a given latitudinal band with the difference in total ET (r of 0.72) than the other two components (vegetation evaporation r of 0.33 and soil evaporation r of 0.19). Similarly, the transpiration difference exhibits a much higher index of agreement with the ET difference than the other two components of ET in the model (0.61, 0.22, and 0.23 for vegetation transpiration, vegetation evaporation, and soil evaporation, respectively; description of index of agreement in Duveiller et al., 2016). We therefore think that focusing on transpiration in a first sensitivity experiment is justified. We encourage however further investigations on the sensitivity of soil and canopy evaporation to land cover and will clarify this in the revised manuscript.

*2) Please provide a metric for quantifying the effects of the modifications. Figure 7 (and the aggregate climate zones) is not adequate for demonstrating significant improvement of the results due to the modifications.*
*Also, while the pft level comparison with GETA looks good, the climate zone compar-*

*ison is more difficult to evaluate. Aggregating to these climate zones smooths out a lot of spatial variability, and may be too coarse to adequately evaluate the modifications. can you calculate a metric to quantify the effects of the modifications? what do pixel-level correlations between the model and the obs look like? are these correlations improved by the modifications? would zonal grouping make more sense than the climate zones?*

**Answer:** We completely agree that introducing a metric will help assessing the performance of different model configurations more objectively. We tested two additional metrics in response to the referee's concern: The root-mean-squared deviation (RMSD) and the index of agreement (IA, as described in Duveiller et al., 2016). For some of the variables with relatively poor agreement (e.g. daily minimum LST difference) the IA tends to be zero or very close to zero in all climate zones. We therefore propose to add a table with the RMSD over the Köppen-Geiger climate zones to the main section of the manuscript and add a table with the IA to the Appendix.

We agree that aggregating over the climate zones smooths out some of the signal. We argue that zonal grouping will create more heterogeneous groups since there can be strong climatic variations at a given latitude as the referee mentions later in his review. We therefore propose to use the following refined climate zones: Equatorial humid (Ef and Em), equatorial seasonally dry (Es and Ew), arid, warm temperate fully humid (Cfa, Cfb, and Cfc), warm temperate summer dry (Csa, Csb, and Csc), warm temperate winter dry (Cwa, Cwb, and Cwc), snow warm summer (Dfa, Dfb, Dsa, Dsb, Dwa, and Dwb), and snow cold summer (Dfc, Dfd, Dsc, Dsd, Dwc, and Dwd). A figure illustrating the global distribution of these climate zones will be added to the Appendix.

*page 6 lines 7-11: I think CLM also outputs a surface radiative temperature. Why didn't you use this?*

**Answer:** To our knowledge, CLM4.5 does not provide a radiative temperature output. Therefore, we added a runtime calculation of radiative temperature in the code which however is calculated according to the CLM4.5 Tech-note (Eq. 4.10 of Oleson et al., 2013).

*page 7 line 21: confidence in which observations? the non-outliers I assume.*

**Answer:** The statement we try to make is that the agreement across different independent data sources gives more confidence in the fact that ET is generally higher over forest. We will reformulate this sentence as: "The considered global ET data sets however consistently exhibit higher ET over forests in most regions (Fig. 2). This agreement across the different independent global data sources gives some confidence in the fact that ET is generally higher over forests."

*page 10 lines 5-6: if comparing for lee et al, why reference alkama and cescatti for the amplification? you should include the delta LST per degree from lee et al for a consistent comparison, and to show that these observations also show this amplification*

**Answer:** It appears that the current formulation of this section creates confusion. Lee et al. (2011) compared temperature measurements 2 m above the canopy to standard station

data. To our knowledge for most of the paired sites there was no direct observations of LST in this study (all but 4). We try to emphasize here that the latitudinal dependence of the T2M difference in CLM4.5 is much weaker than in the observations of Lee et al. (2011) while the latitudinal dependence of LST is slightly stronger than the T2M dependence of Lee et al. (2011). The second part states that we expect a stronger latitudinal dependence of the LST difference compared with the T2M difference since Alkama and Cescatti (2015) observe stronger LST effects of forest in general. We will re-formulate this section to make it more understandable for the reader.

*page 13 line 18: is this because you used prescribed atmospheric forcing?*

**Answer:** We indeed considered this hypothesis (page 9 lines 27-28), but the fact that Lejeune et al. (2017) observed the same for coupled simulations with CLM tends to rule it out.

*Generally, why show a CI for only the modis zonal average? What about the other data and the model outputs? And is CI the best metric to depict variability here? There are many reasons for variability around the globe at a given latitude (e.g., different weather patterns, continental vs maritime), and we should not expect a zonal mean to behave like a population mean estimate that supposedly characterizes a more homogenous group.*

Confidence intervals become very narrow for CLM, GLEAM, and GETA because the zonal sample sizes are much larger. Therefore, they can hardly be seen in Figs. 1 and 5 and were not plotted in Figs. 2 and A9. We agree it is not the ideal metric to display variability here. We will therefore plot the median and the interquartile range instead of the confidence interval in Figs. 1, 2, 5, and A9. An example for such a figure can be seen below for Fig. A9.

[Figure]

**Revised Fig. A9:** Annual mean ET(f-o) in (a) MODIS, (b) GLEAM, (c) GETA, and (d) CLM- BASE. Panel (e) shows the zonal median and the interquartile range of MODIS (in green along with its interquartile range in grey), GLEAM (blue), GETA (orange), and CLM-BASE (red). Note that on this subfigure results have been smoothed with a 4° latitudinally-running mean.

*Figure 2a: this does not appear to be the correct figure. it does not match with the averages in panel 2c, nor table 2*

**Answer:** This seems to be a graphical issue. For some reason there is a thin red line on the margin between data and NaN's which looks dominant when not zooming into the picture. When zoomed in the graph starts to look more blueish. We remade Figs. 2 and A9 using a different format to resolve this (See revised Fig. A9 as an example above).

**Additional remark:** We had to make a correction in the calculation of the mean values over the climate zones for the MODIS data. The resulting changes are small for all climate zones except for the arid zone and do therefore not affect the conclusions of the manuscript (See new figure below).

[Figure]

**Revised Figure 7.** Area-weighted annual mean over Köppen-Geiger climate zones (Kottek et al., 2006) of (a) (f-o), (b) LSTavg(f-o), (c) LSTmax(f-o), and (d) LSTmin(f-o) in MODIS (green), CLM- BASE (red), and CLM-PLUS (orange). Only grid cells containing valid data in the MODIS observations were considered for analysis of CLM4.5. Panel (e) shows the area weighted mean over the Köppen-Geiger climate zone of ET(f-o) in MODIS (green), GLEAM (light blue), GETA (dark blue), CLM- BASE (red), and CLM- PLUS (orange) and panel (f) the area weighted mean ET for each PFT analyzed in this study according to the GETA (dark blue), CLM-BASE (red), and CLM- PLUS (orange). The acronyms of the PFTs are defined in Table 3.

References:

Alkama, R. and Cescatti, A.: Biophysical climate impacts of recent changes in global forest cover, Science, 351, 600–604, 2016.

Duveiller, G., Fasbender, D., and Meroni, M. (2016). Revisiting the concept of a symmetric index of agreement for continuous datasets. Sci. Rep.-UK, 6:19401.

Lee, X., Goulden, M. L., Hollinger, D. Y., Barr, A., Black, T. A., Bohrer, G., Bracho, R., Drake, B., Goldstein, A., Gu, L., Katul, G., Kolb, T., Law, B. E., Margolis, L. H., Meyers, T., Monson, R., Munger, W., Oren, R., Paw U, K. T., Richardson, A. D., Schmid, H. P. Staebler, R.,Wofsy, S., and Zhao, L.: Observed increase in local cooling effect of deforestation at higher latitude, Nature, 479, 384–387, https://doi.org/10.1038/nature10588, 2011.

Lejeune, Q., Seneviratne, S. I., and Davin, E. L.: Historical Land-Cover Change Impacts on Climate: Comparative Assessment of LUCID and CMIP5 Multimodel Experiments, J. Climate, pp. 1439–1459, https://doi.org/10.1175/JCLI-D-16-0213.1, 2017.

Oleson, K. W., Lawrence, D., B. Bonan, G., Drewniak, B., Huang, M., Koven, C., Levis, S., Li, F., Riley, W., M. Subin, Z., C. Swenson, S., E. Thornton, P., Bozbiyik, A., Fisher, R., Heald, C., Kluzek, E., Lamarque, J.-F., Lawrence, P., Leung, L., and Yang, Z.-L.: Technical Description of version 4.5 fo the Community Land Model (CLM), https://doi.org/10.5065/D6RR1W7M, 2013.

---

## Author Response (AR1)

We would like to express our gratitude for the constructive reviews of our manuscript. We have the impression that we greatly improved the quality of our manuscript due to the inputs of the referees. Hereafter we provide an overview of the most important changes in the updated manuscript. After that, we list the comments of referees 1 and 2 in black and a description of the modifications of the manuscript in response to the comment in blue (note that the references made in the responses relate to the unmarked manuscript). After, we display the changes made to the manuscript. Parts that were removed from the manuscript are marked in red and stroke-through, whereas added parts are marked in blue.

There were three major changes to our manuscript:

1) The metrics used to assess the agreement between the model and the observations were calculated over finer climate zones to avoid too heterogeneous groups. The new climate zones are displayed in Fig. 1. The metrics over these climate zones are shown in Figs. 3, 4, and A7.

2) We now evaluate the variables in terms of the root mean squared deviation and the index of agreement (Duveiller et al., 2016), to assess the agreement between the model and the observations more objectively. These data are displayed in Figs. 4 and A7.

3) A Discussion section was added to the manuscript. In the first paragraph, the findings from the Results section are discussed in a broader context. In particular, we focus on how (remaining) biases can further alleviated. The second paragraph discusses why modeled ΔT2M is not in agreement with observations. In the third paragraph we list the limitations of the comparison made in this study. And finally, an outlook on what could be done in future studies of a similar kind is provided in the fourth paragraph.

**Referee 1**

From the stylistic point of view, in my opinion the paper would read better if it is moderately restructured. The description of the sensitivity experiment (section 3.2) should be located in the methodology section before the result. This would avoid the feeling of jumping back and forth from results to methods, and would help justify the presence of CLM-PLUS results in the earlier figures (before its description). I know this may feel awkward as the CLM – PLUS simulation is seen as a response to the problems (i.e. results) identified in section 3.1, but with some effort I am confident the restructuring can be done. I would introduce this idea (that a CLM – PLUS simulation is done as a response to the first results) in the last paragraph of the introduction, and then describe it in the last section of the methods, saying you are anticipating (in the text) the results that will be presented thereafter.

We agree that the experiment description in section 3.2 could be placed elsewhere. We decided to add the methodological details to the appendix dedicated to the sensitivity experiments including the detailed description of the implementation (P16-19). We now focus only on the results in section 3.2 and give a brief overview of the sensitivity experiments in the method section with a reference to the appendix for the more complete details (P4 L25 to P5 L19). As the referee

proposes we describe the overall goal/motivation of the sensitivity experiment in the last part of the introduction (P3 L18-20).

Another restructuring point I would strongly recommend is to try to separate Results from Discussion. The combined section currently works quite well for 'Results', as such a section should not be just a description of results but also an interpretation of them. But some parts can be moved to a more general 'discussion' section in which the whole approach is discussed in a broader sense, providing more insight of the caveats and advantages of the whole experiment, and how it relates to the broader picture in Earth System Science

This is a good suggestion. We added a separate Discussion section in the revised manuscript (P12-14).

Deforestation is more complex than a simple transition from forest to open land described in the Li et al 2015 MODIS dataset, as different types of forest (e.g. evergreen or deciduous) would have different effects (on snow masking and albedo for instance), and different kind of open lands will also behave differently (management would arguably have a strong influence). With the GETA data, the authors do explore this variability for ET to some extent. In my opinion a more thorough discussion is warranted, even if further analyses are not required within this study. Could anything be said on PFT specific differences for albedo and LST? Are there other field-based datasets such as GETA that could be used for these variables? Could other datasets from remote sensing that differentiate amongst forest types be used? If not, mentioning this need could justify and stimulate the development of such products in the future.

We touch on these topics in the newly added Discussion section (P14 L15-20). In particular, it is now mentioned that other land cover conversions appear to be relevant in recently published observational data sets.

There are some doubts on how comparable the deltas that are extracted from GLEAM are with respect to the Li MODIS dataset and to the CLM sub-grid simulations. If I understand correctly, GLEAM provides separate values for tall canopies and low vegetation over the same 0.25 dd pixels, and to obtain a change between 'forest' to 'openland', one makes the difference a pixel level between the value for tall canopies and for low vegetation. However, to understand better the possible repercussions this may have on the analysis, it would be necessary to have more information on how the distinction between tall canopies and low vegetation is made in GLEAM. What land cover maps are used (if any)? How do these match with the CLM distribution of PFTs?

This is an important issue that is now discussed at P13 L31-34.

The MODIS Li et al. 2015 dataset depends on setting a threshold on the percentage of forest/trees that there are in a pixel so as to consider it 'forest' or 'openland'. They also show in their supplementary material that the choice of the threshold does have

some effect of the results. How does this affect the comparison with CLM sub-grid results, for which the signal is fully 'un-mixed'? In my understanding this has the effect that the MODIS delta will often relate to a comparison from a 'not-so-full-forest' to a 'not-so-treeless-openland', while the simulations are from a 'full-closed forest' to a 'treeless openland'. How does this impact the results? Can something be done about it?

We now discuss this in the section where we present the limitations of our evaluation (P13 L34 to P14 L7). Further, newer observation-based data sets resolving this issue are mentioned in the Discussion (P14 L18-22).

Regarding the discussion on T2M vs LST in both the models and observations, an Important point that is not completely clear is whether T2M is considered as 2m above the canopy or above the soil (i.e. within the forest). Note that in studies like Alkama & Cescatti (2016), the techniques to obtain T2M from satellite LST require weather stations, which typically use WMO definitions by which temperature is measured above a standard grass canopy, even if it is surrounded by forest. This means that the T2M obtained is not that which is observed within the canopy (i.e. under the trees) nor the one above the trees. In the model, and hence in this analysis, what temperature are we speaking about and how can the comparability between observations and models be ensured?

T2M in CLM4.5 is defined as the temperature 2 m above the apparent sink for sensible heat (Oleson et al., 2013; Eq. 5.58) which lies within the canopy air space. In the manuscript we argue that T2M is not the right temperature diagnostic to compare to LST observations (this is why we recalculated a radiative temperature (TRAD) based on the outgoing longwave radiation). We nevertheless show a comparison with daily maximum T2M difference in the appendix (Fig. A10) to highlight the different sign of the response in T2M compared to TRAD in CLM4.5. This result is surprising and is worth noting since it implies that modelling studies looking at land use effects might be affected by the choice of temperature diagnostics, which is an issue that has been overlooked in our community. That said, evaluating this T2M temperature signal in CLM4.5 against observations is very challenging since, as the referee rightfully points out, the WMO T2M concept is by definition not applicable to forest and therefore "T2M" in forest is ill-defined. For instance, the measurements of Lee et al. (2011) report "T2M" above the canopy and "T2M" in Alkama and Cescatti (2016) is indeed a mixed concept derived empirically, which is yet another definition compared to the CLM4.5 definition above.
We added these clarifications to the Discussion section (P13 L22-28). Also, we updated Fig. 9 in the manuscript to make it clearer for the reader that we do not intend to compare the T2M signal in CLM with the MODIS LST observations but emphasize that the T2M and the TRAD signal in CLM look very different.

Page 9 Line 31, Could you speculate on why the model would have this behaviour?

We did not further investigate why the model exhibits this behavior. We think that we can exclude the fact that our simulations were made in offline mode as the cause, since online simulations using CLM exhibit similar daily maximum T2M signals (P11 L26-29).

Could you add some info on whether this shallower root distribution
is closer to what is observed, perhaps based on information from the references cited
(Fan2017 & Canadell1996)? Ideally it would be good to have a line in Fig 6 for the
observations over 'openland'.

We now visualize the rooting depths reported in Fan et al. (2017) in Fig. A5.

How do you calculate the confidence interval in MODIS? Do they come from
the original product of Li et al. 2015? If so, do explain a bit more how they are calculated
and how should the reader interpret it?

We replaced the confidence intervals with the interquartile range which is more suitable to
visualize the variability of not-normally-distributed data in Figs. 2, 5, and 8.

For all plots like that of figure 1, I am not too sure how much we gain in insight by
having the fine 1dd resolution. I would recommend using broader latitudinal bins (e.g.
2.5 or perhaps even 5 dd) so as to have larger boxes in which the points of the t-test
are larger and clearer.

At 0.5° resolution the points displaying the results of the t-test were indeed hard to see. We
therefore averaged the CLM and GLEAM data to latitudinal bands of 2.5° in Figs. 2, 6, 7, 8, 9, A4,
and A8.

**Referee 2**

Note that we regrouped the statements of referee 2 regarding the importance of VTR to one
single group in order to avoid repetition in our replies.

1) Complete the analysis of the sources of error. You test only things related to vegeta-
tion transpiration and not soil evaporation. Your data do not clearly indicate that vege-
tation is the main driver, and in fact show that soil evaporation could also be a dominant
source of error. Just because your modifications for transpiration show some improve-
ment does not mean that they are correct, because you could be over-compensating
or over-fitting these parameter values.
page 8 lines 21-22 and 31-32: this statement is not supported by your data or the rest
of this paragraph. while the visual pattern between the VTR and total is similar, the
soil evap effects are compensated for by the interception effects, thus leaving VTR to
dominate the pattern. but this doesn't mean that the soil evap is not a main contributor,
especially in the tropics. and you mention the biases in the non-forest that contribute
to this discrepancy as well. Figure 4 also indicates that the soil evaporation dominates
the total ET pattern in the higher latitudes, which is where your modifications show little
improvement.
page 12 lines 11-22: this indicates that your hypothesis regarding VTR as the main

driver of discrepancies may not be correct. while you get improvements, soil evap remains a problem, and you may even be overcompensating with the VTR related modifications

We agree that the soil and canopy evaporation are both important for the creation of the total ET signal. In the initial manuscript this was not clear enough and the importance of VTR was emphasized too much. Therefore, several statements were reformulated in P10 L3-20. Additionally, it is now mentioned in the Discussion that soil and interception evaporation might contribute considerably to the total ET difference (P13 L11-14).
We assessed the importance of the individual components more objectively by calculating the pearson correlation and the index of agreement (Duveiller et al., 2016) between the monthly difference of forest minus open land of these individual components and the monthly difference in total ET. The difference in vegetation transpiration in CLM4.5 exhibits a stronger correlation over a given latitudinal band with the difference in total ET (r of 0.72) than the other two components (vegetation evaporation r of 0.33 and soil evaporation r of 0.19). Similarly, the transpiration difference exhibits a much higher index of agreement with the ET difference than the other two components of ET in the model (0.61, 0.22, and 0.23 for vegetation transpiration, vegetation evaporation, and soil evaporation, respectively; description of index of agreement in Duveiller et al., 2016).

2) Please provide a metric for quantifying the effects of the modifications. Figure 7 (and the aggregate climate zones) is not adequate for demonstrating significant improvement of the results due to the modifications.
Also, while the pft level comparison with GETA looks good, the climate zone comparison is more difficult to evaluate. Aggregating to these climate zones smooths out a lot of spatial variability, and may be too coarse to adequately evaluate the modifications. can you calculate a metric to quantify the effects of the modifications? what do pixel-level correlations between the model and the obs look like? are these correlations improved by the modifications? would zonal grouping make more sense than the climate zones?

We completely agree that introducing a metric helps assessing the performance of different model configurations more objectively. We tested two additional metrics in response to the referee's concern: The root-mean-squared deviation (RMSD) and the index of agreement (IA, as described in Duveiller et al., 2016). For some of the variables with relatively poor agreement (e.g. daily minimum LST difference) the IA tends to be zero or very close to zero in all climate zones. We therefore added a figure displaying the RMSD over the Köppen-Geiger climate zones to the main section of the manuscript (Fig. 4) and the figure displaying the IA to the appendix (Fig. A7). A description of these metrics was added to the Model Evaluation section (P7 L19 to P8 L4).

We agree that aggregating over the climate zones smooths out some of the signal. We argue that zonal grouping will create more heterogeneous groups since there can be strong climatic variations at a given latitude as the referee mentions later in his review. We therefore calculated the metrics over the following refined climate zones: Equatorial humid (Ef and Em), equatorial seasonally dry (Es and Ew), arid, warm temperate fully humid (Cfa, Cfb, and Cfc), warm temperate summer dry (Csa, Csb, and Csc), warm temperate winter dry (Cwa, Cwb, and Cwc), snow warm summer (Dfa, Dfb, Dsa, Dsb, Dwa, and Dwb), and snow cold summer (Dfc, Dfd, Dsc, Dsd, Dwc, and

Dwd). These new climate zones are displayed in Fig. 1. The values of the different metrics over these climate zones are shown in Figs. 3, 4, and A7.

page 6 lines 7-11: I think CLM also outputs a surface radiative temperature. Why didn't you use this?

To our knowledge, CLM4.5 does not provide a radiative temperature output. Therefore, we added a runtime calculation of radiative temperature in the code which however is calculated according to the CLM4.5 Tech-note (Eq. 4.10 of Oleson et al., 2013) which is described in P7 L7-13.

page 7 line 21: confidence in which observations? the non-outliers I assume.

The statement we try to make is that the agreement across different independent data sources gives more confidence in the fact that ET is generally higher over forest. This section was reformulated (P9 L4-6).

page 10 lines 5-6: if comparing for lee et al, why reference alkama and cescatti for the amplification? you should include the delta LST per degree from lee et al for a consistent comparison, and to show that these observations also show this amplification

The structure and formulation of this section was indeed confusing. The statements regarding T2M were made clearer in the manuscript (P11 L23-29 and P13 L22-29).

page 13 line 18: is this because you used prescribed atmospheric forcing?

We indeed considered this hypothesis. But the fact that Lejeune et al. (2017) observed the same for coupled simulations with CLM tends to rule it out (P11 L26-29).

Generally, why show a CI for only the modis zonal average? What about the other data and the model outputs? And is CI the best metric to depict variability here? There are many reasons for variability around the globe at a given latitude (e.g., different weather patterns, continental vs maritime), and we should not expect a zonal mean to behave like a population mean estimate that supposedly characterizes a more homogenous group.

We agree the confidence interval is not the ideal metric to display variability here. We therefore now plot the mean and the interquartile range instead of the confidence interval in Figs. 2, 5, and 8.

Figure 2a: this does not appear to be the correct figure. it does not match with the averages in panel 2c, nor table 2

This was a graphical issue that has now been corrected for Fig. 5.

[revised manuscript text omitted]